# Stochastic optimization under time drift: iterate averaging, step decay, and high-probability guarantees

**Joshua Cutler**
University of Washington
jocutler@uw.edu

**Dmitriy Drusvyatskiy**
University of Washington
ddrusv@uw.edu

**Zaid Harchaoui**
University of Washington
zaid@uw.edu

## Abstract

We consider the problem of minimizing a convex function that is evolving in time according to unknown and possibly stochastic dynamics. Such problems abound in the machine learning and signal processing literature, under the names of concept drift and stochastic tracking. We provide novel non-asymptotic convergence guarantees for stochastic algorithms with iterate averaging, focusing on bounds valid both in expectation and with high probability. Notably, we show that the tracking efficiency of the proximal stochastic gradient method depends only logarithmically on the initialization quality when equipped with a step-decay schedule.

## 1 Introduction

Stochastic optimization underpins much of machine learning theory and practice. Significant progress has been made over the last two decades in the finite-time analysis of stochastic approximation algorithms; see, e.g., [1, 2, 6, 7, 8, 21, 24, 29, 30]. The predominant assumption in this line of work is that the distribution generating the data is fixed throughout the run of the process. There is no shortage of problems, however, where this assumption is grossly violated for reasons beyond the learner's control. Indeed, data often shifts and evolves over time for reasons that may be independent of the learning process.

Two examples are worth highlighting. The first is a classical problem in signal processing related to stochastic tracking [19, 27], wherein the learning algorithm aims to track over time a moving target driven by an unknown stochastic process. The second example is the concept drift phenomenon in online learning [14, 28], wherein the true hypothesis may be changing over time, as in topic modeling or spam classification. An important goal in online problems, and the one we adopt here, is to track as closely as possible an unknown sequence of minimizers or minimal values. The tracking error efficiency of stochastic algorithms in online settings is much less developed than sample complexity guarantees for static problems.

We present finite-time efficiency estimates in expectation and with high probability for the tracking error of the proximal stochastic gradient method under time drift. Our results concisely explain the interplay between the learning rate, the noise variance in the gradient oracle, and the strength of the time drift. The high-probability results merely assume that the gradient noise and time drift have light tails. Moreover, none of the results require the objectives to have bounded domains. While conventional wisdom and previous work recommend the use of constant step sizes under time drift, we show in an important regime that a significantly better step size schedule is one that is geometrically decaying to a "critical step size".

35th Conference on Neural Information Processing Systems (NeurIPS 2021).

## 1.1 Related work

Our current work fits within the broader literature on stochastic tracking, online optimization with controlled increments, and high-probability guarantees in stochastic optimization. We now survey the most relevant literature in these areas.

**Stochastic tracking.** Stochastic gradient-type algorithms for stochastic tracking and filtering have been the subject of extensive research in the past century. Most works have focused on the so-called least mean-squares (LMS) algorithm and its variants, which can be viewed as a stochastic gradient method on a least-squares loss-based objective. Other stochastic algorithms that have been studied in these settings with a larger cost per iteration include recursive least-squares and the Kalman filter [12]. Recent works have revisited these methods from a more modern viewpoint [5, 23, 32]. In particular, the paper [23] focuses on (accelerated) gradient methods for deterministic tracking problems, while [32] analyzes a stochastic gradient method for online problems that is adaptive to unknown parameters. The paper [5] analyzes the dynamic regret of stochastic algorithms for time-varying problems, focusing both on lower and upper complexity bounds. Though the proof techniques in our paper share many aspects with those available in the literature, the results we obtain are distinct.

**Online optimization with controlled increments.** Online learning under concept drift was first considered by [22] and further developed in several papers [3, 17]. In this literature, the data distribution is typically fixed over time and the rate of variation is stated in terms of the probability of disagreement of consecutive target functions, which is assumed to be upper bounded. Another line of work assumes a time partitioning with an expert in each time interval, and the goal is to compete with the expert in each segment. Closer to this work is [14, 16], where in the framework of online convex optimization the bounds are stated in terms of maximum regret over any contiguous time interval; see also [5, 9, 26]. In contrast to these works, in our framework we state our bounds in the same spirit as in classical stochastic approximation, that is, in terms of distance to optimum and objective function gap, and we present results holding both in expectation and with high probability.

**High-probability guarantees in stochastic optimization.** A large part of our work revolves around high-probability guarantees in stochastic optimization. Classical references on the subject in static settings and for minimizing regret in online optimization include [4, 15, 20, 25]. There exists a variety of techniques for establishing high-probability guarantees based on Freedman's inequality and doubling tricks; see, e.g., [4, 15]. A more recent line of work [13] establishes a generalized Freedman inequality that is custom-tailored for analyzing stochastic gradient-type methods and results in best known high-probability guarantees. Our arguments closely follow the paradigm of [13] based on the generalized Freedman inequality.

## 1.2 Outline

The outline of the paper is as follows. Section 2 formalizes the problem setting of time-dependent stochastic optimization and records the relevant assumptions. Sections 3 and 4 present the main results of the paper. Specifically, Section 3 focuses on efficiency estimates for tracking the minimizer, while Section 4 focuses on efficiency estimates for tracking the minimal value. Proofs of the main results appear in Section 5 of the Supplement, and illustrative numerical results appear in Section 6 of the Supplement.

## 2 Framework and assumptions

### 2.1 Stochastic optimization under time drift

We consider the sequence of stochastic optimization problems

$$\min_x \ \varphi_t(x) := f_t(x) + r_t(x) \tag{1}$$

indexed by time $t \in \mathbb{N}$. We make the standard standing assumption that $(i)$ each function $f_t \colon \mathbb{R}^d \to \mathbb{R}$ is $\mu$-strongly convex and $C^1$-smooth with $L$-Lipschitz continuous gradient for some common parameters $\mu, L > 0$, and $(ii)$ each regularizer $r_t \colon \mathbb{R}^d \to \mathbb{R} \cup \{\infty\}$ is proper, closed and convex. The

minimizer and minimal value of (1) will be denoted by $x_t^\star$ and $\varphi_t^\star$, respectively. Throughout, $\|\cdot\|$ denotes the $\ell_2$-norm on $\mathbb{R}^d$ induced by the dot product $\langle\cdot,\cdot\rangle$.

As motivation, we describe two classical examples of (1) that are worth keeping in mind and that guide our framework: stochastic tracking of a drifting target and online learning under a distributional drift.

**Example 2.1** (Stochastic tracking of a drifting target). The problem of stochastic tracking, related to the filtering problem in signal processing, is to track a moving target $x_t^\star$ from observations
$$b_t = c_t(x_t^\star) + \epsilon_t,$$
where $c_t(\cdot)$ is a known measurement map and $\epsilon_t$ is a mean-zero noise vector. A typical time-dependent problem formulation takes the form
$$\min_x \ \mathbb{E}_{\epsilon_t} \ \ell_t(b_t - c_t(x)) + r_t(x),$$
where $\ell_t(\cdot)$ derives from the distribution of $\epsilon_t$ and $r_t(\cdot)$ encodes available side information about the target $x_t^\star$. Common choices for $r_t$ are the 1-norm and the squared 2-norm. The motion of the target $x_t^\star$ is typically driven by a random walk or a diffusion [12, 27].

**Example 2.2** (Online learning under distributional drift). The problem of online learning under a distributional drift is to learn while the data distribution may change over time. More formally, one problem formulation takes the form
$$\min_x \ \mathbb{E}_{w \sim \mathcal{D}(u_t)} \ \ell(x, w) + r(x).$$
where $\mathcal{D}(u_t)$ is a data distribution that depends on an unknown parameter sequence $\{u_t\}$, which itself may evolve stochastically. The evolution of $u_t$ is often assumed to be piecewise constant in $t$ in online learning [14, 28].

---

**Algorithm 1** Online Proximal Stochastic Gradient $\qquad\qquad\qquad\qquad\qquad$ $\mathtt{PSG}(x_0, \{\eta_t\}, T)$

---

**Input:** initial $x_0$ and step size sequence $\{\eta_t\}_{t=0}^T \subset (0, \infty)$
**Step** $t = 0, \ldots, T-1$:
$$\text{Set } g_t = \widetilde{\nabla} f_t(x_t)$$
$$\text{Set } x_{t+1} = \text{prox}_{\eta_t r_t}(x_t - \eta_t g_t)$$

**Return** $x_T$

---

**Online proximal stochastic gradient method.** The main goal of a learning algorithm for problem (1) is to generate a sequence of points $\{x_t\}$ that minimize some natural performance metric. The most prevalent performance metrics in the literature are the *tracking error* and the *dynamic regret*. We will focus on two types of tracking error, $\|x_t - x_t^\star\|^2$ and $\varphi_t(x_t) - \varphi_t(x_t^\star)$.

We make the standing assumption that at every time $t$, and at every query point $x$, the learner may obtain an *unbiased estimator* $\widetilde{\nabla} f_t(x)$ of the true gradient $\nabla f_t(x)$ in order to proceed with a stochastic gradient-like optimization algorithm. With this oracle access, the online proximal stochastic gradient method—recorded as Algorithm 1 above—in each iteration $t$ simply takes a stochastic gradient step on $f_t$ at $x_t$ followed by a proximal operation on $r_t$:
$$x_{t+1} := \text{prox}_{\eta_t r_t}(x_t - \eta_t g_t) = \underset{u \in \mathbb{R}^d}{\arg\min}\left\{r_t(u) + \frac{1}{2\eta_t}\|u - (x_t - \eta_t g_t)\|^2\right\}.$$

The goal of our work is to obtain efficiency estimates for this procedure that hold both in expectation and with high probability.

**Minimizer drift.** The guarantees we obtain allow both the iterates $x_t$ *and the minimizers* $x_t^\star$ to evolve stochastically. This is convenient for example when tracking a moving target $x_t^\star$ whose motion may be governed by a stochastic process such as a random walk or a diffusion (see Example 2.1). Throughout, we define the *minimizer drift* at time $t$ to be the random variable
$$\Delta_t := \|x_t^\star - x_{t+1}^\star\|.$$

Clearly, an efficiency estimate for Algorithm 1 must take into account the variation of the functions $f_t$ in time $t$. Two of the most popular metrics for measuring such variations are the minimizer drift $\Delta_t$ and the gradient variation $\sup_x \|\nabla f_t(x) - \nabla f_{t+1}(x)\|$. Given identical regularizers, a bound on the gradient variation always implies a bound on the minimizer drift.

**Lemma 2.3** (Gradient variation vs. minimizer drift). *Suppose $i, t \geq 0$ are such that the regularizers $r_i$ and $r_t$ are identical. Then we have*

$$\mu\|x_i^\star - x_t^\star\| \leq \|\nabla f_i(x_t^\star) - \nabla f_t(x_t^\star)\|.$$

## 2.2 Running assumption on the stochastic process

Setting the stage, given $\{x_t\}$ and $\{g_t\}$ as in Algorithm 1 we let

$$z_t := \nabla f_t(x_t) - g_t$$

denote the *gradient noise* at time $t$ and we impose the following assumption modeling stochasticity in the online problem throughout Sections 3 and 4.

**Assumption 2.4** (Stochastic framework). There exists a filtered probability space $(\Omega, \mathcal{F}, \mathbb{F}, \mathbb{P})$ with filtration $\mathbb{F} = (\mathcal{F}_t)_{t \geq 0}$ such that $\mathcal{F}_0 = \{\emptyset, \Omega\}$ and the following holds for all $t \geq 0$:

  (i) $x_t, x_t^\star \colon \Omega \to \mathbb{R}^d$ are $\mathcal{F}_t$-measurable,

  (ii) $z_t \colon \Omega \to \mathbb{R}^d$ is $\mathcal{F}_{t+1}$-measurable with $\mathbb{E}[z_t \mid \mathcal{F}_t] = 0$.

The first item of Assumption 2.4 simply says that $x_t$ and $x_t^\star$ are fully determined by information up to time $t$. The second item of Assumption 2.4 asserts that the gradient noise $z_t$ is fully determined by information up to time $t + 1$ and has zero mean conditioned on the information up to time $t$; for example, this holds naturally in Example 2.2 if we take $g_t = \nabla \ell(x_t, w_t)$ with $w_t \sim \mathcal{D}(u_t)$ provided the loss $\ell(\cdot, w_t)$ is $C^1$-smooth.

# 3 Tracking the minimizer with the last iterate

In this section, we present bounds on the tracking error $\|x_t - x_t^\star\|^2$ that are valid both in expectation and with high probability under light-tail assumptions. Further, we show that a geometrically decaying learning rate schedule may be superior to a constant learning rate in terms of efficiency.

## 3.1 Bounds in expectation

We begin with bounding the expected value $\mathbb{E}\|x_t - x_t^\star\|^2$. The starting point for our analysis is the following standard one-step improvement guarantee.

**Lemma 3.1** (One-step improvement). *For all $x \in \mathbb{R}^d$, the iterates $\{x_t\}$ produced by Algorithm 1 with $\eta_t < 1/L$ satisfy the bound:*

$$2\eta_t(\varphi_t(x_{t+1}) - \varphi_t(x)) \leq (1 - \mu\eta_t)\|x_t - x\|^2 - \|x_{t+1} - x\|^2 + 2\eta_t\langle z_t, x_t - x\rangle + \frac{\eta_t^2}{1 - L\eta_t}\|z_t\|^2.$$

For simplicity, we state the main results under the assumption that the second moments $\mathbb{E}\big[\Delta_t^2\big]$ and $\mathbb{E}\|z_t\|^2$ are uniformly bounded; more general guarantees that take into account weighted averages of the moments and allow for time-dependent learning rates follow from Lemma 3.1 as well.

**Assumption 3.2** (Bounded second moments). There exist constants $\Delta, \sigma > 0$ such that the following holds for all $t \geq 0$.

  (i) **(Drift)** The minimizer drift $\Delta_t$ satisfies $\mathbb{E}\big[\Delta_t^2\big] \leq \Delta^2$.

  (ii) **(Noise)** The gradient noise $z_t$ satisfies $\mathbb{E}\|z_t\|^2 \leq \sigma^2$.

The following theorem establishes an expected improvement guarantee for Algorithm 1, and serves as the basis for much of what follows; see Section 5.1 of the Supplement for the precise statements and proofs of the present section.

**Theorem 3.3** (Expected distance). *Suppose that Assumption 3.2 holds. Then the iterates produced by Algorithm 1 with constant learning rate $\eta \leq 1/2L$ satisfy the bound:*

$$\mathbb{E}\|x_t - x_t^\star\|^2 \lesssim \underbrace{(1 - \mu\eta)^t \|x_0 - x_0^\star\|^2}_{optimization} + \underbrace{\frac{\eta\sigma^2}{\mu}}_{noise} + \underbrace{\left(\frac{\Delta}{\mu\eta}\right)^2}_{drift}.$$

**Interplay of optimization, noise, and drift.** Theorem 3.3 states that when using a constant learning rate, the error $\mathbb{E}\|x_t - x_t^\star\|^2$ decays linearly in time $t$, until it reaches the "noise+drift" error $\eta\sigma^2/\mu + (\Delta/\mu\eta)^2$. Notice that the "noise+drift" error cannot be made arbitrarily small. This is perfectly in line with intuition: a learning rate that is too small prevents the algorithm from catching up with $x_t^\star$. We note that the individual error terms due to the optimization and noise are classically known to be tight for PSG; tightness of the drift term is proved in [23, Theorem 3.2].

With Theorem 3.3 in hand, we are led to define the following asymptotic tracking error of Algorithm 1 corresponding to $\mathbb{E}\|x_t - x_t^\star\|^2$, together with the corresponding optimal step size:

$$\mathcal{E} := \min_{\eta \in (0, 1/2L]} \left\{ \frac{\eta\sigma^2}{\mu} + \left(\frac{\Delta}{\mu\eta}\right)^2 \right\} \quad \text{and} \quad \eta_\star := \min\left\{ \frac{1}{2L}, \left(\frac{2\Delta^2}{\mu\sigma^2}\right)^{1/3} \right\}.$$

Plugging $\eta_\star$ into the definition of $\mathcal{E}$, we see that Algorithm 1 exhibits qualitatively different behaviors in settings corresponding to high or low drift-to-noise ratio $\Delta/\sigma$, explicitly given by

$$\mathcal{E} \asymp \begin{cases} \frac{\sigma^2}{\mu L} + \left(\frac{L\Delta}{\mu}\right)^2 & \text{if } \frac{\Delta}{\sigma} \geq \sqrt{\frac{\mu}{16L^3}} \\ \left(\frac{\Delta\sigma^2}{\mu^2}\right)^{2/3} & \text{otherwise.} \end{cases}$$

Two regimes of variation are brought to light by the above computation: the *high drift-to-noise regime* $\Delta/\sigma \geq \sqrt{\mu/16L^3}$, and the *low drift-to-noise regime* $\Delta/\sigma < \sqrt{\mu/16L^3}$. The high drift-to-noise regime is uninteresting from the viewpoint of stochastic optimization because the optimal learning rate is as large as in the deterministic setting, $\eta_\star = 1/2L$. In contrast, the low drift-to-noise regime is interesting because the optimal learning rate $\eta_\star = (2\Delta^2/\mu\sigma^2)^{1/3}$ is smaller than $1/2L$ and exhibits a nontrivial scaling with the problem parameters.

**Learning rate vs. rate of variation.** A central question is to find a learning rate schedule that achieves a tracking error $\mathbb{E}\|x_t - x_t^\star\|^2$ that is within a constant factor of $\mathcal{E}$ in the shortest possible time. The answer is clear in the high drift-to-noise regime $\Delta/\sigma \geq \sqrt{\mu/16L^3}$. Indeed, in this case, Theorem 3.3 directly implies that Algorithm 1 with the constant learning rate $\eta_\star = 1/2L$ will find a point $x_t$ satisfying $\mathbb{E}\|x_t - x_t^\star\|^2 \lesssim \mathcal{E}$ in time $t \lesssim (L/\mu) \log(\|x_0 - x_0^\star\|^2/\mathcal{E})$. Notice that the efficiency estimate is logarithmic in $1/\mathcal{E}$; intuitively, the reason for the absence of a sublinear component is that the error due to the drift $\Delta$ dominates the error due to the variance $\sigma^2$ in the stochastic gradient.

The low drift-to-noise regime $\Delta/\sigma < \sqrt{\mu/16L^3}$ is more subtle. Namely, the simplest strategy is to execute Algorithm 1 with the constant learning rate $\eta_\star = (2\Delta^2/\mu\sigma^2)^{1/3}$. Then a direct application of Theorem 3.3 yields the estimate $\mathbb{E}\|x_t - x_t^\star\|^2 \lesssim \mathcal{E}$ in time $t \lesssim (\sigma^2/\mu^2\mathcal{E}) \log(\|x_0 - x_0^\star\|^2/\mathcal{E})$. This efficiency estimate can be significantly improved by gradually decaying the learning rate using a "step-decay schedule", wherein the algorithm is implemented in epochs with the new learning rate chosen to be the midpoint between the current learning rate and $\eta_\star$. Such schedules are well known to improve efficiency in the static setting, as was discovered in [10, 11], and can be used here. The end result is the following theorem (see Theorem 5.5 of the Supplement for the precise statement).

**Theorem 3.4** (Time to track in expectation, informal). *Suppose that Assumption 3.2 holds. Then there is a learning rate schedule $\{\eta_t\}$ such that Algorithm 1 produces a point $x_t$ satisfying*

$$\mathbb{E}\|x_t - x_t^\star\|^2 \lesssim \mathcal{E} \quad \text{after time} \quad t \lesssim \frac{L}{\mu} \log\left(\frac{\|x_0 - x_0^\star\|^2}{\mathcal{E}}\right) + \frac{\sigma^2}{\mu^2\mathcal{E}}.$$

Remarkably, the efficiency estimate in Theorem 3.4 looks identical to the efficiency estimate in the classical static setting [20], with $\mathcal{E}$ playing the role of the target accuracy $\varepsilon$. Theorems 3.3 and 3.4 provide useful baseline guarantees for the performance of Algorithm 1. Nonetheless, these guarantees

are all stated in terms of the *expected* tracking error $\mathbb{E}\|x_t - x_t^\star\|^2$, and are therefore only meaningful if the entire algorithm can be repeated from scratch multiple times. There is no shortage of situations in which a learning algorithm is operating in real time and the time drift is irreversible; in such settings, the algorithm may only be executed once. Such settings call for efficiency estimates that hold with high probability, rather than only in expectation.

## 3.2 High-probability guarantees

We next present high-probability guarantees on the tracking error $\|x_t - x_t^\star\|^2$. To this end, we make the following standard light-tail assumptions on the minimizer drift and gradient noise [13, 20, 24].

**Assumption 3.5** (Sub-Gaussian drift and noise). *There exist constants $\Delta, \sigma > 0$ such that the following holds for all $t \geq 0$.*

(i) **(Drift)** *The square drift $\Delta_t^2$ is sub-exponential conditioned on $\mathcal{F}_t$ with parameter $\Delta^2$:*
$$\mathbb{E}\big[\exp(\lambda \Delta_t^2)\,|\,\mathcal{F}_t\big] \leq \exp(\lambda \Delta^2) \quad \text{for all} \quad 0 \leq \lambda \leq \Delta^{-2}.$$

(ii) **(Noise)** *The noise $z_t$ is norm sub-Gaussian conditioned on $\mathcal{F}_t$ with parameter $\sigma/2$:*
$$\mathbb{P}\big\{\|z_t\| \geq \tau \,|\, \mathcal{F}_t\big\} \leq 2\exp(-2\tau^2/\sigma^2) \quad \text{for all} \quad \tau > 0.$$

Note that the first item of Assumption 3.5 is equivalent to asserting that the minimizer drift $\Delta_t$ is sub-Gaussian conditioned on $\mathcal{F}_t$. Clearly Assumption 3.5 implies Assumption 3.2 with the same constants $\Delta, \sigma$. It is worthwhile to note some common settings in which Assumption 3.5 holds; the claims in Remark 3.6 follow from standard results on sub-Gaussian random variables [18, 31].

**Remark 3.6** (Common settings for Assumption 3.5). Fix constants $\Delta, \sigma > 0$. If $\Delta_t$ is bounded by $\Delta$, then clearly $\Delta_t^2$ is sub-exponential (conditioned on $\mathcal{F}_t$) with parameter $\Delta^2$. Similarly, if $\|z_t\|$ is bounded by $\sigma$, then $z_t$ is norm sub-Gaussian (conditioned on $\mathcal{F}_t$) with parameter $\sigma/2$ (by Markov's inequality). Alternatively, if the increment $x_t^\star - x_{t+1}^\star$ is mean-zero sub-Gaussian conditioned on $\mathcal{F}_t$ with parameter $\Delta/\sqrt{d}$, then $x_t^\star - x_{t+1}^\star$ is mean-zero norm sub-Gaussian conditioned on $\mathcal{F}_t$ with parameter $2\sqrt{2} \cdot \Delta$ and hence $\Delta_t^2$ is sub-exponential conditioned on $\mathcal{F}_t$ with parameter $c \cdot \Delta^2$ for some absolute constant $c > 0$. Similarly, if $z_t$ is sub-Gaussian conditioned on $\mathcal{F}_t$ with parameter $\sigma/4\sqrt{2d}$, then $z_t$ is norm sub-Gaussian conditioned on $\mathcal{F}_t$ with parameter $\sigma/2$.

The following theorem shows that if Assumption 3.5 holds, then the expected bound on $\|x_t - x_t^\star\|^2$ derived in Theorem 3.3 holds with high probability.

**Theorem 3.7** (High-probability distance tracking). *Suppose that Assumption 3.5 holds and let $\{x_t\}$ be the iterates produced by Algorithm 1 with constant learning rate $\eta \leq 1/2L$. Then there is an absolute constant $c > 0$ such that for any specified $t \in \mathbb{N}$ and $\delta \in (0, 1)$, the estimate*

$$\|x_t - x_t^\star\|^2 \leq \left(1 - \tfrac{\mu\eta}{2}\right)^t \|x_0 - x_0^\star\|^2 + c\left(\frac{\eta\sigma^2}{\mu} + \left(\frac{\Delta}{\mu\eta}\right)^2\right)\log\left(\frac{e}{\delta}\right)$$

*holds with probability at least $1 - \delta$.*

The proof of Theorem 3.7 employs a technique used in [13]. The main idea is to build a careful recursion for the moment generating function of $\|x_t - x_t^\star\|^2$, leading to a one-sided sub-exponential tail bound. As a consequence of Theorem 3.7, we can again implement a step-decay schedule in the low drift-to-noise regime to obtain the following efficiency estimate with high probability; see Section 5.2 of the Supplement for the formal statements and proofs.

**Theorem 3.8** (Time to track with high probability, informal). *Suppose that Assumption 3.5 holds and that we are in the low drift-to-noise regime $\Delta/\sigma < \sqrt{\mu/16L^3}$. Then there is a learning rate schedule $\{\eta_t\}$ such that for any specified $\delta \in (0, 1)$, Algorithm 1 produces a point $x_t$ satisfying*

$$\|x_t - x_t^\star\|^2 \lesssim \mathcal{E}\log\left(\frac{e}{\delta}\right)$$

*with probability at least $1 - K\delta$ after time*

$$t \lesssim \frac{L}{\mu} \log\left(\frac{\|x_0 - x_0^\star\|^2}{\mathcal{E}}\right) + \frac{\sigma^2}{\mu^2 \mathcal{E}}, \quad \text{where } K \lesssim \log_2\left(\frac{1}{L} \cdot \left(\frac{\sigma^2 \mu}{\Delta^2}\right)^{1/3}\right).$$

## 4 Tracking the minimal value

The results outlined so far have focused on tracking the minimizer $x_t^\star$. In this section, we present results for tracking the minimal value $\varphi_t^\star$. These two goals are fundamentally different. Generally speaking, good bounds on the function gap along with strong convexity imply good bounds on the distance to the minimizer; the reverse implication is false. To this end, we require a stronger assumption on the variation of the functions $f_t$ in time $t$: rather than merely controlling the minimizer drift $\Delta_t$, we will assume control on the *gradient drift*

$$G_{i,t} := \sup_x \|\nabla f_i(x) - \nabla f_t(x)\|.$$

Our strategy is to track the minimal value along a running average $\hat{x}_t$ of the iterates $x_t$ produced by Algorithm 1, as defined in Algorithm 2 below. The reason behind using this particular running average is brought to light in Section 5.3 of the Supplement, where we apply a standard averaging technique to a one-step improvement along $x_t$ to obtain the desired progress along $\hat{x}_t$.

---

**Algorithm 2** Averaged Online Proximal Stochastic Gradient $\qquad\qquad \overline{\mathsf{PSG}}(x_0, \mu, \{\eta_t\}, T)$

---

**Input:** initial $x_0 =: \hat{x}_0$, strong convexity parameter $\mu$, and step size sequence $\{\eta_t\}_{t=0}^T \subset (0, 2\mu^{-1})$
**Step** $t = 0, \ldots, T - 1$:

$$\text{Set } g_t = \widetilde{\nabla} f_t(x_t)$$
$$\text{Set } x_{t+1} = \text{prox}_{\eta_t r_t}(x_t - \eta_t g_t)$$
$$\text{Set } \hat{x}_{t+1} = \left(1 - \frac{\mu\eta_t}{2 - \mu\eta_t}\right)\hat{x}_t + \frac{\mu\eta_t}{2 - \mu\eta_t} x_{t+1}$$

**Return** $\hat{x}_T$

---

### 4.1 Bounds in expectation

We begin with bounding the expected value $\mathbb{E}[\varphi_t(\hat{x}_t) - \varphi_t^\star]$. Analogous to Assumption 3.2, we make the following assumption regarding drift and noise.

**Assumption 4.1** (Bounded second moments)**.** The regularizers $r_t \equiv r$ are identical for all times $t$ and there exist constants $\Delta, \sigma > 0$ such that the following properties hold for all $0 \leq i < t$:

(i) **(Drift)** The gradient drift $G_{i,t}$ satisfies $\mathbb{E}\big[G_{i,t}^2\big] \leq (\mu\Delta|i - t|)^2$.

(ii) **(Noise)** The gradient noise $z_i$ satisfies $\mathbb{E}\|z_i\|^2 \leq \sigma^2$ and $\mathbb{E}\langle z_i, x_t^\star \rangle = 0$.

These two assumptions are natural indeed. Taking into account Lemma 2.3, it is clear that Assumption 4.1 implies the earlier Assumption 3.2 with the same constants $\Delta, \sigma$. The assumption on the drift intuitively asserts that gradient drift $G_{i,t}$ can grow only linearly in time $|i - t|$ (in expectation). In particular, returning to Example 2.2, suppose that the distribution map $\mathcal{D}(\cdot)$ is $\gamma$-Lipschitz continuous in the Wasserstein-1 distance, the loss $\ell(\cdot, w)$ is $C^1$ smooth for all $w$, and the gradient $\nabla\ell(x, \cdot)$ is $\beta$-Lipschitz continuous for all $x$. Then the Kantorovich-Rubinstein duality theorem directly implies $\mathbb{E}\big[G_{i,t}^2\big] \leq (\gamma\beta)^2 \mathbb{E}\|u_i - u_t\|^2$. Therefore, as long as the second moment $\mathbb{E}\|u_i - u_t\|^2$ scales quadratically in $|i - t|$, the desired drift assumption holds. The assumption on the noise requires a uniform bound on the second moment $\mathbb{E}\|z_i\|^2$ and for the condition $\mathbb{E}\langle z_i, x_t^\star \rangle = 0$ to hold. The latter property confers a weak form of uncorrelatedness between the gradient noise $z_i$ and the future minimizer $x_t^\star$, and holds automatically if the gradient noise and the minimizers evolve independently of each other, as would typically be the case for instance in Example 2.2.

The following theorem provides an expected improvement guarantee for Algorithm 2; for the full statement and proof, see Corollary 5.12 of the Supplement.

**Theorem 4.2** (Expected function gap). *Suppose that Assumption 4.1 holds, and let $\{\hat{x}_t\}$ be the iterates produced by Algorithm 2 with constant learning rate $\eta \leq 1/2L$. Then the following bound holds for all $t \geq 0$:*

$$\mathbb{E}\big[\varphi_t(\hat{x}_t) - \varphi_t^\star\big] \lesssim \underbrace{\Big(1 - \tfrac{\mu\eta}{2}\Big)^t \big(\varphi_0(x_0) - \varphi_0^\star\big)}_{optimization} + \underbrace{\eta\sigma^2}_{noise} + \underbrace{\frac{\Delta^2}{\mu\eta^2}}_{drift}.$$

The "noise+drift" error term in Theorem 4.2 coincides with $\mu$ times the corresponding error term in Theorem 3.3, as expected due to $\mu$-strong convexity. With Theorem 4.2 in hand, we are led to define the following asymptotic tracking error of Algorithm 2 corresponding to $\mathbb{E}[\varphi_t(\hat{x}_t) - \varphi_t^\star]$:

$$\mathcal{G} := \mu\mathcal{E} = \min_{\eta \in (0, 1/2L]} \left\{ \eta\sigma^2 + \frac{\Delta^2}{\mu\eta^2} \right\}.$$

The corresponding asymptotically optimal choice of $\eta$ is again given by $\eta_\star$, and the dichotomy governed by the drift-to-noise ratio $\Delta/\sigma$ remains:

$$\mathcal{G} \asymp \begin{cases} \frac{\sigma^2}{L} + \frac{(L\Delta)^2}{\mu} & \text{if } \frac{\Delta}{\sigma} \geq \sqrt{\frac{\mu}{16L^3}} \\ \mu\left(\frac{\Delta\sigma^2}{\mu^2}\right)^{2/3} & \text{otherwise.} \end{cases}$$

In the high drift-to-noise regime $\Delta/\sigma \geq \sqrt{\mu/16L^3}$, Theorem 4.2 directly implies that Algorithm 2 with the constant learning rate $\eta_\star = 1/2L$ finds a point $\hat{x}_t$ satisfying $\mathbb{E}[\varphi_t(\hat{x}_t) - \varphi_t^\star] \lesssim \mathcal{G}$ in time $t \lesssim (L/\mu) \log((\varphi_0(x_0) - \varphi_0^\star)/\mathcal{G})$. In the low drift-to-noise regime $\Delta/\sigma < \sqrt{\mu/16L^3}$, another direct application of Theorem 4.2 shows that Algorithm 2 with the constant learning rate $\eta_\star = (2\Delta^2/\mu\sigma^2)^{1/3}$ finds a point $\hat{x}_t$ satisfying $\mathbb{E}[\varphi_t(\hat{x}_t) - \varphi_t^\star] \lesssim \mathcal{G}$ in time $t \lesssim (\sigma^2/\mu\mathcal{G}) \log((\varphi_0(x_0) - \varphi_0^\star)/\mathcal{G})$. As before, this efficiency estimate can be significantly improved by implementing a step-decay schedule. The end result is the following theorem; see Theorem 5.13 of the Supplement for the formal statement and proof.

**Theorem 4.3** (Time to track in expectation, informal). *Suppose that Assumption 4.1 holds. Then there is a learning rate schedule $\{\eta_t\}$ such that Algorithm 2 produces a point $\hat{x}_t$ satisfying*

$$\mathbb{E}[\varphi_t(\hat{x}_t) - \varphi_t^\star] \lesssim \mathcal{G} \quad \text{after time} \quad t \lesssim \frac{L}{\mu} \log\left(\frac{\varphi_0(x_0) - \varphi_0^\star}{\mathcal{G}}\right) + \frac{\sigma^2}{\mu\mathcal{G}}.$$

## 4.2 High-probability guarantees

Our next result is an analogue of Theorem 4.2 that holds with high probability. Naturally, such a result should rely on light-tail assumptions on the gradient drift $G_{i,t}$ and the norm of the gradient noise $\|z_i\|$. We state the guarantee under the assumption that $G_{i,t}$ and $\|z_i\|$ are conditionally sub-Gaussian (Assumption 4.4). In particular, we require for the first time that the gradient noise $z_i$ is mean-zero conditioned on the $\sigma$-algebra

$$\mathcal{F}_{i,t} := \sigma(\mathcal{F}_i, x_t^\star)$$

for all $0 \leq i < t$; the property $\mathbb{E}[z_i \mid \mathcal{F}_{i,t}] = 0$ would follow from independence of the gradient noise $z_i$ and the future minimizer $x_t^\star$ and is reasonable in light of Examples 2.1 and 2.2.

**Assumption 4.4** (Sub-Gaussian drift and noise). *The regularizers $r_t \equiv r$ are identical for all times $t$ and there exist constants $\Delta, \sigma > 0$ such that the following properties hold for all $0 \leq i < t$.*

(i) **(Drift)** *The square gradient drift $G_{i,t}^2$ is sub-exponential with parameter $(\mu\Delta|i - t|)^2$:*

$$\mathbb{E}\big[\exp\big(\lambda G_{i,t}^2\big)\big] \leq \exp\big(\lambda(\mu\Delta|i - t|)^2\big) \quad \text{for all} \quad 0 \leq \lambda \leq (\mu\Delta|i - t|)^{-2}.$$

(ii) **(Noise)** *The gradient noise $z_i$ is mean-zero norm sub-Gaussian conditioned on $\mathcal{F}_{i,t}$ with parameter $\sigma/2$, i.e., $\mathbb{E}[z_i \mid \mathcal{F}_{i,t}] = 0$ and*

$$\mathbb{P}\big\{\|z_i\| \geq \tau \mid \mathcal{F}_{i,t}\big\} \leq 2\exp(-2\tau^2/\sigma^2) \quad \text{for all} \quad \tau \geq 0.$$

Clearly the chain of implications holds:

$$\text{Assumption 4.4} \implies \text{Assumption 4.1} \implies \text{Assumption 3.2}.$$

**Example 4.5** (Sub-Gaussian feature model). In the setting of logistic regression, sub-Gaussian gradient noise naturally arises from sampling from a sub-Gaussian feature model. Indeed, in this case the objective takes the form $f(x) = \mathbb{E}_{A,b}\left[\sum_{i=1}^n \log(1 + \exp\langle a_i, x\rangle) - \langle Ax, b\rangle\right]$ and drawing $(A, b)$ yields the sample gradient $\widetilde{\nabla} f(x) = A^T(S(Ax) - b)$, where $A \in \mathbb{R}^{n \times d}$ has rows $a_1, ..., a_n \in \mathbb{R}^d$ and $S$ denotes the sigmoid function. Being that $S$ and $b$ are bounded, it therefore follows that if the rows of $A$ are sub-Gaussian, then so is the gradient noise $\nabla f(x) - \widetilde{\nabla} f(x)$.

The following theorem shows that if Assumption 4.4 holds, then the expected bound on $\varphi_t(\hat{x}_t) - \varphi_t^\star$ derived in Theorem 4.2 holds with high probability.

**Theorem 4.6** (Function gap with high probability). *Suppose that Assumption 4.4 holds, and let $\{\hat{x}_t\}$ be the iterates produced by Algorithm 2 with constant learning rate $\eta \leq 1/2L$. Then there is an absolute constant $c > 0$ such that for any specified $t \in \mathbb{N}$ and $\delta \in (0, 1)$, the estimate*

$$\varphi_t(\hat{x}_t) - \varphi_t^\star \leq c\left(\left(1 - \tfrac{\mu\eta}{2}\right)^t\left(\varphi_0(x_0) - \varphi_0^\star\right) + \eta\sigma^2 + \frac{\Delta^2}{\mu\eta^2}\right)\log\left(\frac{e}{\delta}\right)$$

*holds with probability at least $1 - \delta$.*

The proof of Theorem 4.6 is based on combining the generalized Freedman inequality of [13] with careful control on the drift and noise in improvement guarantees for the proximal stochastic gradient method. The key observation is that although we do not have simple recursive control on the moment generating function of $\varphi_t(\hat{x}_t) - \varphi_t^\star$ (as we do with $\|x_t - x_t^\star\|^2$), we *can* control the tracking error $\varphi_t(\hat{x}_t) - \varphi_t^\star$ by leveraging control on the martingale $\sum_{i=0}^{t-1}\langle z_i, x_i - x_t^\star\rangle\zeta^{t-1-i}$, where $\zeta = 1 - \mu\eta/(2 - \mu\eta)$. This martingale is self-regulating in the sense that its total conditional variance is bound by the history of the process; the generalized Freedman inequality is precisely suited to bound such martingales with high probability.

With Theorem 4.6 in hand, we may implement a step-decay schedule as before to obtain the following efficiency estimate; see Section 5.4 of the Supplement for the formal statements and proofs.

**Theorem 4.7** (Time to track with high probability, informal). *Suppose that Assumption 4.4 holds and that we are in the low drift-to-noise regime $\Delta/\sigma < \sqrt{\mu/16L^3}$. Fix $\delta \in (0, 1)$. Then there is a learning rate schedule $\{\eta_t\}$ such that Algorithm 2 produces a point $\hat{x}_t$ satisfying*

$$\varphi_t(\hat{x}_t) - \varphi_t^\star \lesssim \mathcal{G}\log\left(\frac{e}{\delta}\right)$$

*with probability at least $1 - K\delta$ after time*

$$t \lesssim \frac{L}{\mu}\log\left(\frac{\varphi_0(x_0) - \varphi_0^\star}{\mathcal{G}}\right) + \frac{\sigma^2}{\mu\mathcal{G}}\log\left(\log\left(\frac{e}{\delta}\right)\right), \quad \text{where } K \lesssim \log_2\left(\frac{1}{L}\cdot\left(\frac{\sigma^2\mu}{\Delta^2}\right)^{1/3}\right).$$

# 5 Conclusion

We presented finite-time efficiency estimates in expectation and with high probability for the tracking error of the proximal stochastic gradient method under time drift. Through our investigation of the interplay between the learning rate, the noise variance in the gradient oracle, and the strength of the time drift, we uncovered two regimes of interest, each suggesting a different choice of learning rate. The high-probability guarantees extend recent results to the time-dependent setting.

## Acknowledgments

This work was supported by NSF CCF-2019844, NSF DMS-2023166, NSF DMS-1651851, CIFAR-LMB, and faculty research awards. Part of this work was done while Z. Harchaoui was visiting the Simons Institute for the Theory of Computing.

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
