# 5 Proofs of results in Sections 3 and 4

**Roadmap.** Throughout this section, we enforce the assumptions and notation of Section 2 and let $\{x_t\}$ denote the iterates generated by Algorithm 1 with $\eta_t < 1/L$. Sections 5.1 and 5.2 handle distance tracking under time drift: Section 5.1 derives the results of Section 3.1, while Section 5.2 derives the results of Section 3.2. Then Sections 5.3 and 5.4 handle function gap tracking under time drift: Section 5.3 derives the results of Section 4.1, while Section 5.4 derives the results of Section 4.2.

## 5.1 Tracking the minimizer: bounds in expectation

The proof of Theorem 3.3 follows a familiar pattern in stochastic optimization. We begin by recalling Lemma 3.1, which gives a standard one-step improvement guarantee [22] for the proximal stochastic gradient method on the fixed problem $\min \varphi_t$.

**Lemma 5.1** (One-step improvement). *The estimate*

$$2\eta_t(\varphi_t(x_{t+1}) - \varphi_t(x)) \le (1 - \mu\eta_t)\|x_t - x\|^2 - \|x_{t+1} - x\|^2 + 2\eta_t\langle z_t, x_t - x\rangle + \tfrac{\eta_t^2}{1 - L\eta_t}\|z_t\|^2$$

*holds for all points $x \in \mathbb{R}^d$ and for all indices $t \ge 0$.*

*Proof.* Since $f_t$ is $L$-smooth, we have

$$
\begin{aligned}
\varphi_t(x_{t+1}) &= f_t(x_{t+1}) + r_t(x_{t+1}) \\
&\le f_t(x_t) + \langle \nabla f_t(x_t), x_{t+1} - x_t\rangle + \tfrac{L}{2}\|x_{t+1} - x_t\|^2 + r_t(x_{t+1}) \\
&= f_t(x_t) + r_t(x_{t+1}) + \langle g_t, x_{t+1} - x_t\rangle + \tfrac{L}{2}\|x_{t+1} - x_t\|^2 + \langle z_t, x_{t+1} - x_t\rangle.
\end{aligned}
$$

Next, given any $\delta_t > 0$, Young's inequality yields

$$\langle z_t, x_{t+1} - x_t\rangle \le \tfrac{\delta_t}{2}\|z_t\|^2 + \tfrac{1}{2\delta_t}\|x_{t+1} - x_t\|^2.$$

Therefore, given any $x \in \mathbb{R}^d$, we have

$$
\begin{aligned}
\varphi_t(x_{t+1}) &\le f_t(x_t) + r_t(x_{t+1}) + \langle g_t, x_{t+1} - x_t\rangle + \tfrac{\delta_t^{-1} + L}{2}\|x_{t+1} - x_t\|^2 + \tfrac{\delta_t}{2}\|z_t\|^2 \\
&= f_t(x_t) + r_t(x_{t+1}) + \langle g_t, x_{t+1} - x_t\rangle + \tfrac{1}{2\eta_t}\|x_{t+1} - x_t\|^2 \\
&\quad + \tfrac{\delta_t^{-1} + L - \eta_t^{-1}}{2}\|x_{t+1} - x_t\|^2 + \tfrac{\delta_t}{2}\|z_t\|^2 \\
&\le f_t(x_t) + r_t(x) + \langle g_t, x - x_t\rangle + \tfrac{1}{2\eta_t}\|x - x_t\|^2 - \tfrac{1}{2\eta_t}\|x - x_{t+1}\|^2 \\
&\quad + \tfrac{\delta_t^{-1} + L - \eta_t^{-1}}{2}\|x_{t+1} - x_t\|^2 + \tfrac{\delta_t}{2}\|z_t\|^2,
\end{aligned}
$$

where the last inequality holds because $x_{t+1} = \operatorname{prox}_{\eta_t r_t}(x_t - \eta_t g_t)$ is the minimizer of the $\eta_t^{-1}$-strongly convex function $r_t + \langle g_t, \cdot - x_t\rangle + \tfrac{1}{2\eta_t}\|\cdot - x_t\|^2$. Now we estimate

$$
\begin{aligned}
f_t(x_t) + r_t(x) + \langle g_t, x - x_t\rangle &= f_t(x_t) + \langle \nabla f_t(x_t), x - x_t\rangle + r_t(x) + \langle z_t, x_t - x\rangle \\
&\le f_t(x) - \tfrac{\mu}{2}\|x - x_t\|^2 + r_t(x) + \langle z_t, x_t - x\rangle \\
&= \varphi_t(x) - \tfrac{\mu}{2}\|x - x_t\|^2 + \langle z_t, x_t - x\rangle
\end{aligned}
$$

using the $\mu$-strong convexity of $f_t$. Thus,

$$
\begin{aligned}
\varphi_t(x_{t+1}) \le \varphi_t(x) - \tfrac{\mu}{2}\|x - x_t\|^2 + \langle z_t, x_t - x\rangle + \tfrac{1}{2\eta_t}\|x - x_t\|^2 - \tfrac{1}{2\eta_t}\|x - x_{t+1}\|^2 \\
+ \tfrac{\delta_t^{-1} + L - \eta_t^{-1}}{2}\|x_{t+1} - x_t\|^2 + \tfrac{\delta_t}{2}\|z_t\|^2.
\end{aligned}
$$

Finally, taking $\delta_t = \eta_t/(1 - L\eta_t)$ and rearranging (note that $\varphi_t(x_{t+1})$ is finite) yields

$$2\eta_t(\varphi_t(x_{t+1}) - \varphi_t(x)) \le (1 - \mu\eta_t)\|x_t - x\|^2 - \|x_{t+1} - x\|^2 + 2\eta_t\langle z_t, x_t - x\rangle + \tfrac{\eta_t^2}{1 - L\eta_t}\|z_t\|^2,$$

as claimed. $\qquad\square$

It is critically important that the one-step improvement estimate in Lemma 5.1 holds with respect to any reference point $x$. In particular, setting $x = x_t^\star$ yields the following lemma.

**Lemma 5.2** (Distance recursion). *The estimate*

$$\|x_{t+1} - x_{t+1}^\star\|^2 \le (1 - \mu\eta_t)\|x_t - x_t^\star\|^2 + 2\eta_t\langle z_t, x_t - x_t^\star\rangle + \frac{\eta_t^2}{1-L\eta_t}\|z_t\|^2 + \left(1 + \frac{1}{\mu\eta_t}\right)\Delta_t^2$$

*holds for all indices $t \ge 0$.*

*Proof.* Note that the $\mu$-strong convexity of $\varphi_t$ implies $\frac{\mu}{2}\|x_{t+1} - x_t^\star\|^2 \le \varphi_t(x_{t+1}) - \varphi_t^\star$. Combining this estimate with Lemma 5.1 under the identification $x = x_t^\star$ yields

$$(1 + \mu\eta_t)\|x_{t+1} - x_t^\star\|^2 \le (1 - \mu\eta_t)\|x_t - x_t^\star\|^2 + 2\eta_t\langle z_t, x_t - x_t^\star\rangle + \frac{\eta_t^2}{1-L\eta_t}\|z_t\|^2.$$

Next, an application of Young's inequality reveals

$$\|x_{t+1} - x_{t+1}^\star\|^2 \le (1 + \mu\eta_t)\|x_{t+1} - x_t^\star\|^2 + (1 + (\mu\eta_t)^{-1})\|x_t^\star - x_{t+1}^\star\|^2,$$

thereby completing the proof. $\square$

Applying Lemma 5.2 recursively furnishes a bound on $\|x_t - x_t^\star\|^2$. When the step size is constant, the next proposition follows immediately.

**Proposition 5.3** (Last-iterate progress). *Suppose $\eta_t \equiv \eta$. Then the following bound holds for all $t \ge 0$:*

$$\|x_t - x_t^\star\|^2 \le (1 - \mu\eta)^t\|x_0 - x_0^\star\|^2 + 2\eta\sum_{i=0}^{t-1}\langle z_i, x_i - x_i^\star\rangle(1 - \mu\eta)^{t-1-i}$$

$$+ \frac{\eta^2}{1-L\eta}\sum_{i=0}^{t-1}\|z_i\|^2(1 - \mu\eta)^{t-1-i} + \left(1 + \frac{1}{\mu\eta}\right)\sum_{i=0}^{t-1}\Delta_i^2(1 - \mu\eta)^{t-1-i}.$$

By taking expectations in Proposition 5.3, we obtain the following precise version of Theorem 3.3.

**Corollary 5.4** (Expected distance). *Suppose that Assumption 3.2 holds. Then the iterates $\{x_t\}$ generated by Algorithm 1 with constant learning rate $\eta \le 1/2L$ satisfy the bound:*

$$\mathbb{E}\|x_t - x_t^\star\|^2 \le (1 - \mu\eta)^t\|x_0 - x_0^\star\|^2 + 2\left(\frac{\eta\sigma^2}{\mu} + \left(\frac{\Delta}{\mu\eta}\right)^2\right).$$

With Corollary 5.4 in hand, we can now prove an expected efficiency estimate for the online proximal stochastic gradient method using a step-decay schedule, wherein the algorithm is implemented in epochs with the new learning rate chosen to be the midpoint between the current learning rate and $\eta_\star$. The following is the formal statement of Theorem 3.4 (as previously noted, in the high drift-to-noise regime $\Delta/\sigma \ge \sqrt{\mu/16L^3}$, Theorem 3.4 holds trivially with the constant learning rate $\eta_\star = 1/2L$). The argument is close in spirit to the justifications of the restart schemes in [11, 12].

**Theorem 5.5** (Time to track in expectation). *Suppose that Assumption 3.2 holds and that we are in the low drift-to-noise regime $\Delta/\sigma < \sqrt{\mu/16L^3}$. Set $\eta_\star = (2\Delta^2/\mu\sigma^2)^{1/3}$ and $\mathcal{E} = (\Delta\sigma^2/\mu^2)^{2/3}$. Suppose moreover that we have available a positive upper bound on the initial square distance $D \ge \|x_0 - x_0^\star\|^2$. Consider running Algorithm 1 in $k = 0, \dots, K - 1$ epochs, namely, set $X_0 = x_0$ and iterate the process*

$$X_{k+1} = \mathtt{PSG}(X_k, \eta_k, T_k) \quad for \quad k = 0, \dots, K - 1,$$

*where the number of epochs is*

$$K = 1 + \left\lceil \log_2\left(\frac{1}{L} \cdot \left(\frac{\sigma^2\mu}{\Delta^2}\right)^{1/3}\right)\right\rceil$$

*and we set*

$$\eta_0 = \frac{1}{2L}, \quad T_0 = \left\lceil \frac{2L}{\mu}\log\left(\frac{\mu LD}{\sigma^2}\right)^+\right\rceil \quad and \quad \eta_k = \frac{\eta_{k-1} + \eta_\star}{2}, \quad T_k = \left\lceil \frac{\log(4)}{\mu\eta_k}\right\rceil \quad \forall k \ge 1.$$

*Then the time horizon $T = T_0 + \cdots + T_{K-1}$ satisfies*

$$T \lesssim \frac{L}{\mu} \log \left( \frac{\mu L D}{\sigma^2} \right)^+ + \frac{\sigma^2}{\mu^2 \mathcal{E}} \leq \frac{L}{\mu} \log \left( \frac{D}{\mathcal{E}} \right)^+ + \frac{\sigma^2}{\mu^2 \mathcal{E}},$$

*while the corresponding tracking error satisfies $\mathbb{E}\|X_K - X_K^\star\|^2 \lesssim \mathcal{E}$, where $X_K^\star$ denotes the minimizer of $\varphi_T$.*

*Proof.* For each index $k$, let $t_k := T_0 + \cdots + T_{k-1}$ (with $t_0 := 0$), $X_k^\star$ be the minimizer of the corresponding function $\varphi_{t_k}$, and

$$E_k := \frac{2}{\mu} \left( \eta_k \sigma^2 + \frac{\Delta^2}{\mu \eta_\star^2} \right).$$

Then taking into account $\eta_k \geq \eta_\star$, Corollary 5.4 directly implies

$$\mathbb{E}\|X_{k+1} - X_{k+1}^\star\|^2 \leq (1 - \mu\eta_k)^{T_k} \mathbb{E}\|X_k - X_k^\star\|^2 + \frac{2}{\mu} \left( \eta_k \sigma^2 + \frac{\Delta^2}{\mu \eta_k^2} \right)$$

$$\leq e^{-\mu\eta_k T_k} \mathbb{E}\|X_k - X_k^\star\|^2 + E_k.$$

We will verify by induction that the estimate $\mathbb{E}\|X_{k+1} - X_{k+1}^\star\|^2 \leq 2E_k$ holds for all indices $k$. To see the base case, observe

$$\mathbb{E}\|X_1 - X_1^\star\|^2 \leq e^{-\mu\eta_0 T_0} \|X_0 - X_0^\star\|^2 + E_0 \leq 2E_0.$$

Assume next that the claim holds for index $k - 1$. We then conclude

$$\mathbb{E}\|X_{k+1} - X_{k+1}^\star\|^2 \leq e^{-\mu\eta_k T_k} \mathbb{E}\|X_k - X_k^\star\|^2 + E_k$$

$$\leq \frac{1}{4} \mathbb{E}\|X_k - X_k^\star\|^2 + E_k \leq \frac{E_k}{2E_{k-1}} \mathbb{E}\|X_k - X_k^\star\|^2 + E_k \leq 2E_k,$$

thereby completing the induction. Hence $\mathbb{E}\|X_K - X_K^\star\|^2 \leq 2E_{K-1}$. Next, observe

$$E_{K-1} - \sqrt[3]{54} \left( \frac{\Delta\sigma^2}{\mu^2} \right)^{2/3} = \frac{2\sigma^2}{\mu}(\eta_{K-1} - \eta_\star) = \frac{2\sigma^2}{\mu} \cdot \frac{\eta_0 - \eta_\star}{2^{K-1}} \leq \left( \frac{\Delta\sigma^2}{\mu^2} \right)^{2/3},$$

so

$$\mathbb{E}\|X_K - X_K^\star\|^2 \leq 2(1 + \sqrt[3]{54}) \left( \frac{\Delta\sigma^2}{\mu^2} \right)^{2/3} \asymp \mathcal{E}.$$

Finally, note

$$T \lesssim \frac{L}{\mu} \log \left( \frac{\mu L D}{\sigma^2} \right)^+ + \frac{1}{\mu} \sum_{k=1}^{K-1} \frac{1}{\eta_k}$$

and

$$\sum_{k=1}^{K-1} \frac{1}{\eta_k} \leq 2L \sum_{k=1}^{K-1} 2^k \leq 2L \cdot 2^K = 8L \cdot 2^{K-2} \leq 8 \left( \frac{\sigma^2 \mu}{\Delta^2} \right)^{1/3} = \frac{8\sigma^2}{\mu} \cdot \left( \frac{\Delta\sigma^2}{\mu^2} \right)^{-2/3} \asymp \frac{\sigma^2}{\mu \mathcal{E}}.$$

This completes the proof. $\square$

## 5.2 Tracking the minimizer: high-probability guarantees

The proof strategy of Theorem 3.7 follows a similar argument as in [14, Claim D.1], which recursively controls the moment generating function of $\|x_t - x_t^\star\|^2$. Namely, Lemma 5.2 in the regime $\eta_t \leq 1/2L$ directly yields

$$\|x_{t+1} - x_{t+1}^\star\|^2 \leq (1 - \mu\eta_t)\|x_t - x_t^\star\|^2 + 2\eta_t \langle z_t, u_t \rangle \|x_t - x_t^\star\| + 2\eta_t^2 \|z_t\|^2 + \frac{2}{\mu\eta_t} \Delta_t^2, \quad (4)$$

where we set $u_t := \frac{x_t - x_t^\star}{\|x_t - x_t^\star\|}$ if $x_t$ is distinct from $x_t^\star$ and set it to zero otherwise. The right-hand side has the form of a contraction factor, gradient noise, and drift. The goal is now to control the moment generating function $\mathbb{E}[e^{\lambda\|x_t - x_t^\star\|^2}]$ through this recursion. The basic probabilistic tool for similar

settings under bounded noise assumptions was developed in [14]. The following proposition is a slight generalization of [14, Claim D.1] to a light-tail setting.

**Proposition 5.6** (Recursive control on MGF). *Consider scalar stochastic processes $(V_t)$, $(D_t)$, and $(X_t)$ on a probability space with filtration $(\mathcal{H}_t)$, which are linked by the inequality*

$$V_{t+1} \leq \alpha_t V_t + D_t \sqrt{V_t} + X_t + \kappa_t$$

*for some deterministic constants $\alpha_t \in (-\infty, 1]$ and $\kappa_t \in \mathbb{R}$. Suppose the following properties hold.*

- *$V_t$ is nonnegative and $\mathcal{H}_t$-measurable.*

- *$D_t$ is mean-zero sub-Gaussian conditioned on $\mathcal{H}_t$ with deterministic parameter $\sigma_t$:*
$$\mathbb{E}[\exp(\lambda D_t) \,|\, \mathcal{H}_t] \leq \exp(\lambda^2 \sigma_t^2 / 2) \quad \text{for all} \quad \lambda \in \mathbb{R}.$$

- *$X_t$ is nonnegative and sub-exponential conditioned on $\mathcal{H}_t$ with deterministic parameter $\nu_t$:*
$$\mathbb{E}[\exp(\lambda X_t) \,|\, \mathcal{H}_t] \leq \exp(\lambda \nu_t) \quad \text{for all} \quad 0 \leq \lambda \leq 1/\nu_t.$$

*Then the estimate*

$$\mathbb{E}[\exp(\lambda V_{t+1})] \leq \exp\left(\lambda(\nu_t + \kappa_t)\right) \mathbb{E}\left[\exp\left(\lambda\left(\frac{1+\alpha_t}{2}\right) V_t\right)\right]$$

*holds for any $\lambda$ satisfying $0 \leq \lambda \leq \min\left\{\frac{1-\alpha_t}{2\sigma_t^2}, \frac{1}{2\nu_t}\right\}$.*

*Proof.* For any index $t$ and any scalar $\lambda \geq 0$, the tower rule implies

$$\mathbb{E}[\exp(\lambda V_{t+1})] \leq \mathbb{E}\left[\exp\left(\lambda\left(\alpha_t V_t + D_t \sqrt{V_t} + X_t + \kappa_t\right)\right)\right]$$
$$= \exp(\lambda \kappa_t) \mathbb{E}\left[\exp(\lambda \alpha_t V_t) \mathbb{E}\left[\exp\left(\lambda D_t \sqrt{V_t}\right) \exp(\lambda X_t) \,|\, \mathcal{H}_t\right]\right].$$

Hölder's inequality in turn yields

$$\mathbb{E}\left[\exp\left(\lambda D_t \sqrt{V_t}\right) \exp(\lambda X_t) \,|\, \mathcal{H}_t\right] \leq \sqrt{\mathbb{E}\left[\exp\left(2\lambda \sqrt{V_t} D_t\right) \,|\, \mathcal{H}_t\right] \cdot \mathbb{E}[\exp(2\lambda X_t) \,|\, \mathcal{H}_t]}$$
$$\leq \sqrt{\exp(2\lambda^2 V_t \sigma_t^2) \exp(2\lambda \nu_t)}$$
$$= \exp(\lambda^2 \sigma_t^2 V_t) \exp(\lambda \nu_t)$$

provided $0 \leq \lambda \leq \frac{1}{2\nu_t}$. Thus, if $0 \leq \lambda \leq \min\left\{\frac{1-\alpha_t}{2\sigma_t^2}, \frac{1}{2\nu_t}\right\}$, then the following estimate holds:

$$\mathbb{E}[\exp(\lambda V_{t+1})] \leq \exp(\lambda \kappa_t) \mathbb{E}\left[\exp(\lambda \alpha_t V_t) \exp(\lambda^2 \sigma_t^2 V_t) \exp(\lambda \nu_t)\right]$$
$$= \exp\left(\lambda(\nu_t + \kappa_t)\right) \mathbb{E}\left[\exp(\lambda(\alpha_t + \lambda \sigma_t^2) V_t)\right]$$
$$\leq \exp\left(\lambda(\nu_t + \kappa_t)\right) \mathbb{E}\left[\exp\left(\lambda\left(\frac{1+\alpha_t}{2}\right) V_t\right)\right].$$

The proof is complete. $\qquad\square$

We may now use Proposition 5.6 to derive the following precise version of Theorem 3.7.

**Theorem 5.7** (High-probability distance tracking). *Suppose that Assumption 3.5 holds and let $\{x_t\}$ be the iterates produced by Algorithm 1 with constant learning rate $\eta \leq 1/2L$. Then there exists an absolute constant[1] $c > 0$ such that for any specified $t \in \mathbb{N}$ and $\delta \in (0, 1)$, the estimate*

$$\|x_t - x_t^\star\|^2 \leq \left(1 - \frac{\mu\eta}{2}\right)^t \|x_0 - x_0^\star\|^2 + \left(\frac{8\eta(c\sigma)^2}{\mu} + 4\left(\frac{\Delta}{\mu\eta}\right)^2\right) \log\left(\frac{e}{\delta}\right)$$

*holds with probability at least $1 - \delta$.*

---

[1]Explicitly, one can take any $c \geq 1$ such that $\|z_t\|^2$ is sub-exponential conditioned on $\mathcal{F}_t$ with parameter $c\sigma^2$ and $z_t$ is mean-zero sub-Gaussian conditioned on $\mathcal{F}_t$ with parameter $c\sigma$ for all $t$.

*Proof.* Note first that under Assumption 3.5, there exists an absolute constant $c \geq 1$ such that $\|z_t\|^2$ is sub-exponential conditioned on $\mathcal{F}_t$ with parameter $c\sigma^2$ and $z_t$ is mean-zero sub-Gaussian conditioned on $\mathcal{F}_t$ with parameter $c\sigma$ for all $t$. Therefore $\langle z_t, u_t \rangle$ is mean-zero sub-Gaussian conditioned on $\mathcal{F}_t$ with parameter $c\sigma$, while $\Delta_t^2$ is sub-exponential conditioned on $\mathcal{F}_t$ with parameter $\Delta^2$ by assumption. Thus, in light of inequality (4), we may apply Proposition 5.6 with $\mathcal{H}_t = \mathcal{F}_t$, $V_t = \|x_t - x_t^\star\|^2$, $D_t = 2\eta_t \langle z_t, u_t \rangle$, $X_t = 2\eta_t^2 \|z_t\|^2 + 2\Delta_t^2/\mu\eta_t$, $\alpha_t = 1 - \mu\eta_t$, $\kappa_t = 0$, $\sigma_t = 2\eta_t c\sigma$, and $\nu_t = 2\eta_t^2 c\sigma^2 + 2\Delta^2/\mu\eta_t$, yielding the estimate

$$\mathbb{E}\left[\exp\left(\lambda \|x_{t+1} - x_{t+1}^\star\|^2\right)\right] \leq \exp\left(\lambda\left(2\eta_t^2 c\sigma^2 + \frac{2\Delta^2}{\mu\eta_t}\right)\right) \mathbb{E}\left[\exp\left(\lambda\left(1 - \tfrac{\mu\eta_t}{2}\right)\|x_t - x_t^\star\|^2\right)\right]$$

(5)

for all

$$0 \leq \lambda \leq \min\left\{\frac{\mu}{8\eta_t(c\sigma)^2}, \frac{1}{4\eta_t^2 c\sigma^2 + 4\Delta^2/\mu\eta_t}\right\}.$$

Taking into account $\eta_t \equiv \eta$ and iterating the recursion (5), we deduce

$$\mathbb{E}\left[\exp\left(\lambda \|x_t - x_t^\star\|^2\right)\right] \leq \exp\left(\lambda\left(1 - \tfrac{\mu\eta}{2}\right)^t \|x_0 - x_0^\star\|^2 + \lambda\left(2\eta^2 c\sigma^2 + \frac{2\Delta^2}{\mu\eta}\right)\sum_{i=0}^{t-1}\left(1 - \tfrac{\mu\eta}{2}\right)^i\right)$$

$$\leq \exp\left(\lambda\left(\left(1 - \tfrac{\mu\eta}{2}\right)^t \|x_0 - x_0^\star\|^2 + \frac{4\eta c\sigma^2}{\mu} + 4\left(\frac{\Delta}{\mu\eta}\right)^2\right)\right)$$

for all

$$0 \leq \lambda \leq \min\left\{\frac{\mu}{8\eta(c\sigma)^2}, \frac{1}{4\eta^2 c\sigma^2 + 4\Delta^2/\mu\eta}\right\}.$$

Moreover, setting

$$\nu := \frac{8\eta(c\sigma)^2}{\mu} + 4\left(\frac{\Delta}{\mu\eta}\right)^2$$

and taking into account $c \geq 1$ and $\mu\eta \leq 1$, we have

$$\frac{4\eta c\sigma^2}{\mu} + 4\left(\frac{\Delta}{\mu\eta}\right)^2 \leq \nu$$

and

$$\frac{1}{\nu} = \frac{\mu}{8\eta(c\sigma)^2 + 4\Delta^2/\mu\eta^2} \leq \min\left\{\frac{\mu}{8\eta(c\sigma)^2}, \frac{1}{4\eta^2 c\sigma^2 + 4\Delta^2/\mu\eta}\right\}.$$

Hence

$$\mathbb{E}\left[\exp\left(\lambda\left(\|x_t - x_t^\star\|^2 - \left(1 - \tfrac{\mu\eta}{2}\right)^t \|x_0 - x_0^\star\|^2\right)\right)\right] \leq \exp(\lambda\nu) \quad \text{for all} \quad 0 \leq \lambda \leq 1/\nu.$$

Taking $\lambda = 1/\nu$ and applying Markov's inequality completes the proof. $\qquad\square$

With Theorem 3.7 in hand, we can now prove a high-probability efficiency estimate for the online proximal stochastic gradient method using a step-decay schedule. The following theorem is the precise form of Theorem 3.8. The argument follows the same reasoning as in the proof of Theorem 5.5, with Theorem 3.7 playing the role of Corollary 5.4 while using a union bound over the epochs. The proof appears in the appendix (see Section A.1).

**Theorem 5.8** (Time to track with high probability). *Suppose that Assumption 3.5 holds and that we are in the low drift-to-noise regime $\Delta/\sigma < \sqrt{\mu/16L^3}$. Set $\eta_\star = (2\Delta^2/\mu\sigma^2)^{1/3}$ and $\mathcal{E} = (\Delta\sigma^2/\mu^2)^{2/3}$. Suppose moreover that we have available an upper bound on the initial square distance $D \geq \|x_0 - x_0^\star\|^2$. Consider running Algorithm 1 in $k = 0, \ldots, K-1$ epochs, namely, set $X_0 = x_0$ and iterate the process*

$$X_{k+1} = \mathtt{PSG}(X_k, \eta_k, T_k) \quad \text{for} \quad k = 0, \ldots, K-1,$$

*where the number of epochs is*

$$K = 1 + \left\lceil \log_2\left(\frac{1}{L} \cdot \left(\frac{\sigma^2\mu}{\Delta^2}\right)^{1/3}\right)\right\rceil$$

*and we set*

$$\eta_0 = \frac{1}{2L}, \quad T_0 = \left\lceil \frac{4L}{\mu} \log \left( \frac{\mu L D}{\sigma^2} \right)^+ \right\rceil \quad and \quad \eta_k = \frac{\eta_{k-1} + \eta_\star}{2}, \quad T_k = \left\lceil \frac{2 \log(4)}{\mu \eta_k} \right\rceil \qquad \forall k \geq 1.$$

*Then the time horizon $T = T_0 + \cdots + T_{K-1}$ satisfies*

$$T \lesssim \frac{L}{\mu} \log \left( \frac{\mu L D}{\sigma^2} \right)^+ + \frac{\sigma^2}{\mu^2 \mathcal{E}} \leq \frac{L}{\mu} \log \left( \frac{D}{\mathcal{E}} \right)^+ + \frac{\sigma^2}{\mu^2 \mathcal{E}},$$

*and for any specified $\delta \in (0,1)$, the corresponding tracking error satisfies*

$$\|X_K - X_K^\star\|^2 \lesssim \mathcal{E} \log \left( \frac{e}{\delta} \right)$$

*with probability at least $1 - K\delta$, where $X_K^\star$ denotes the minimizer of $\varphi_T$.*

## 5.3 Tracking the minimal value: bounds in expectation

We turn now to tracking the minimal value. Henceforth, we suppose $\eta_t \leq 1/2L$ and that the regularizers $r_t \equiv r$ are identical for all times $t$. Setting the stage, fix a time horizon $t$. Then Lemma 5.1 directly yields the following one-step improvement guarantee for all indices $i$:

$$2\eta_i(\varphi_i(x_{i+1}) - \varphi_i(x_t^\star)) \leq (1 - \mu\eta_i)\|x_i - x_t^\star\|^2 - \|x_{i+1} - x_t^\star\|^2 + 2\eta_i\langle z_i, x_i - x_t^\star\rangle + 2\eta_i^2\|z_i\|^2.$$

Notice that this provides an estimate on the "wrong quantity" $\varphi_i(x_{i+1}) - \varphi_i(x_t^\star)$, whereas we would like to obtain an estimate on the suboptimality gap $\varphi_t(x_{i+1}) - \varphi_t(x_t^\star)$. In words, we would like to replace $\varphi_i$ with $\varphi_t$, while the controlling the incurred error. Lemma 5.9 shows that the incurred error can by controlled by the gradient drift $G_{i,t}$, while Lemma 5.10 deduces the desired one-step improvement guarantee on $\varphi_t$.

**Lemma 5.9** (Gradient drift vs. gap variation). *For all indices $i, t \in \mathbb{N}$ and points $x, y \in \operatorname{dom} r$, the estimate holds:*

$$\left| [\varphi_i(y) - \varphi_i(x)] - [\varphi_t(y) - \varphi_t(x)] \right| \leq G_{i,t}\|y - x\|.$$

*Proof.* Taking into account $r_t \equiv r$ and using the fundamental theorem of calculus, we may write

$$[\varphi_i(y) - \varphi_i(x)] - [\varphi_t(y) - \varphi_t(x)] = \int_0^1 \langle \nabla f_i(x + s(y-x)) - \nabla f_t(x + s(y-x)), y - x \rangle \, ds$$
$$\leq G_{i,t}\|y - x\|,$$

where the last inequality follows from Cauchy-Schwarz. Switching $x$ and $y$ completes the proof. □

**Lemma 5.10** (One-step improvement). *For all indices $i, t \in \mathbb{N}$, points $x \in \operatorname{dom} r$, and arbitrary $\alpha > 0$, we have*

$$2\eta_i(\varphi_t(x_{i+1}) - \varphi_t(x)) \leq (1 - \mu\eta_i)\|x_i - x\|^2 - (1 - \alpha\eta_i)\|x_{i+1} - x\|^2$$
$$+ 2\eta_i\langle z_i, x_i - x\rangle + 2\eta_i^2\|z_i\|^2 + \frac{\eta_i}{\alpha}G_{i,t}^2.$$

*Proof.* This follows immediately from combining Lemmas 5.1 and 5.9 and Young's inequality, $2G_{i,t}\|x_{i+1} - x\| \leq \alpha^{-1}G_{i,t}^2 + \alpha\|x_{i+1} - x\|^2$. □

Turning the estimate in Lemma 5.10 into an efficiency guarantee on the average iterate is essentially standard and follows for example from the averaging techniques in [10, 11, 12, 20]. The resulting progress along the average iterate is summarized in the following proposition, while the description of the key averaging lemma is placed in the appendix (see Section A.2).

**Proposition 5.11** (Progress along the average iterate). *Let $\{\hat{x}_t\}$ be the iterates produced by Algorithm 2 with constant step size $\eta \leq 1/2L$; thus, setting $\hat{\rho} := \mu\eta/(2 - \mu\eta)$, we have $\hat{x}_0 = x_0$ and*

$\hat{x}_t = (1 - \hat{\rho})\,\hat{x}_{t-1} + \hat{\rho}x_t$ *for all* $t \geq 1$. *Then the following bound holds for all* $t \geq 0$ *and* $x \in \mathrm{dom}\, r$:

$$\varphi_t(\hat{x}_t) - \varphi_t(x) \leq (1 - \hat{\rho})^t\left(\varphi_t(x_0) - \varphi_t(x) + \tfrac{\mu}{4}\|x_0 - x\|^2\right) + \hat{\rho}\sum_{i=0}^{t-1}\langle z_i, x_i - x\rangle(1 - \hat{\rho})^{t-1-i}$$

$$+ \hat{\rho}\eta\sum_{i=0}^{t-1}\|z_i\|^2(1 - \hat{\rho})^{t-1-i} + \tfrac{\hat{\rho}}{\mu}\sum_{i=0}^{t-1}G_{i,t}^2(1 - \hat{\rho})^{t-1-i}.$$

*Proof.* Setting $\alpha = \mu/2$ in Lemma 5.10, we obtain the following recursion for all indices $k \geq 0$ and $t \geq 1$ and points $x \in \mathrm{dom}\, r$:

$$\rho(\varphi_k(x_t) - \varphi_k(x)) \leq (1 - c_1\rho)V_{t-1} - (1 + c_2\rho)V_t + \omega_t,$$

where $\rho = 2\eta$, $c_1 = \mu/2$, $c_2 = -\mu/4$, $V_i = \|x_i - x\|^2$, and $\omega_t = 2\eta\langle z_{t-1}, x_{t-1} - x\rangle + 2\eta^2\|z_{t-1}\|^2 + (2\eta/\mu)G_{t-1,k}^2$. The result follows by applying the averaging Lemma A.1 with $h = \varphi_t - \varphi_t(x)$. $\quad\square$

Taking expectations in Proposition 5.11, we obtain the following precise version of Theorem 4.2.

**Corollary 5.12** (Expected function gap)**.** *Suppose that Assumption 4.1 holds, let* $\{\hat{x}_t\}$ *be the iterates produced by Algorithm 2 with constant step size* $\eta \leq 1/2L$, *and set* $\hat{\rho} := \mu\eta/(2 - \mu\eta)$. *Then the following bound holds for all* $t \geq 0$:

$$\mathbb{E}\big[\varphi_t(\hat{x}_t) - \varphi_t^\star\big] \leq (1 - \hat{\rho})^t \cdot \mathbb{E}\big[\varphi_t(x_0) - \varphi_t^\star + \tfrac{\mu}{4}\|x_0 - x_t^\star\|^2\big] + \eta\sigma^2 + \frac{8\Delta^2}{\mu\eta^2}. \qquad (6)$$

*Consequently, we have*

$$\mathbb{E}\big[\varphi_t(\hat{x}_t) - \varphi_t^\star\big] \lesssim (1 - \hat{\rho})^t\big(\varphi_0(x_0) - \varphi_0^\star\big) + \eta\sigma^2 + \frac{\Delta^2}{\mu\eta^2}$$

*for all* $t \geq 0$, *and the following asymptotic error bound holds:*

$$\limsup_{t\to\infty}\mathbb{E}\big[\varphi_t(\hat{x}_t) - \varphi_t^\star\big] \leq \eta\sigma^2 + \frac{8\Delta^2}{\mu\eta^2}.$$

*Proof.* The bound (6) follows by setting $x = x_t^\star$ in Proposition 5.11, taking expectations, and noting

$$\sum_{i=0}^{t-1}\mathbb{E}\|z_i\|^2(1 - \hat{\rho})^{t-1-i} \leq \frac{\sigma^2}{\hat{\rho}} \qquad \text{and} \qquad \sum_{i=0}^{t-1}\mathbb{E}\big[G_{i,t}^2\big](1 - \hat{\rho})^{t-1-i} \leq \frac{(\mu\Delta)^2(2 - \hat{\rho})}{\hat{\rho}^3}$$

by Assumption 4.1. Next, applying Lemma 5.9 and Young's inequality together with the $\mu$-strong convexity of $\varphi_0$ and Lemma 2.3 yields

$$\varphi_t(x_0) - \varphi_t^\star + \tfrac{\mu}{4}\|x_0 - x_t^\star\|^2 \leq 3\big(\varphi_0(x_0) - \varphi_0^\star\big) + 2\mu^{-1}G_{0,t}^2, \qquad (7)$$

and then taking expectations and invoking Assumption 4.1 gives

$$\mathbb{E}\big[\varphi_t(x_0) - \varphi_t^\star + \tfrac{\mu}{4}\|x_0 - x_t^\star\|^2\big] \leq 3\big(\varphi_0(x_0) - \varphi_0^\star\big) + 2\mu\Delta^2 t^2. \qquad (8)$$

Further, the inequality

$$e^{-\mu\eta t/2}\mu t^2 \leq 16/\mu\eta^2 \qquad \forall \mu, \eta, t > 0 \qquad (9)$$

combines with inequality (8) to yield

$$(1 - \hat{\rho})^t \cdot \mathbb{E}\big[\varphi_t(x_0) - \varphi_t^\star + \tfrac{\mu}{4}\|x_0 - x_t^\star\|^2\big] \leq 3(1 - \hat{\rho})^t\big(\varphi_0(x_0) - \varphi_0^\star\big) + \frac{32\Delta^2}{\mu\eta^2}$$

and the remaining assertions of the corollary follow. $\quad\square$

We may now apply Corollary 5.12 to obtain the formal version of Theorem 4.3; the proof closely follows that of Theorem 5.5 and is included in the appendix (see Section A.3).

**Theorem 5.13** (Time to track in expectation)**.** *Suppose that Assumption 4.1 holds and that we are in the low drift-to-noise regime* $\Delta/\sigma < \sqrt{\mu/16L^3}$. *Set* $\eta_\star = (2\Delta^2/\mu\sigma^2)^{1/3}$ *and* $\mathcal{G} = \mu(\Delta\sigma^2/\mu^2)^{2/3}$. *Suppose moreover that we have available a positive upper bound on the initial gap* $D \geq \varphi_0(x_0) - \varphi_0^\star$.

Consider running Algorithm 2 in $k = 0, \ldots, K - 1$ epochs, namely, set $X_0 = x_0$ and iterate the process

$$X_{k+1} = \overline{\mathtt{PSG}}(X_k, \mu, \eta_k, T_k) \quad for \quad k = 0, \ldots, K - 1,$$

where the number of epochs is

$$K = 1 + \left\lceil \log_2 \left( \frac{1}{L} \cdot \left( \frac{\sigma^2 \mu}{\Delta^2} \right)^{1/3} \right) \right\rceil$$

and we set

$$\eta_0 = \frac{1}{2L}, \quad T_0 = \left\lceil \frac{4L}{\mu} \log \left( \frac{LD}{\sigma^2} \right)^+ \right\rceil \quad and \quad \eta_k = \frac{\eta_{k-1} + \eta_\star}{2}, \quad T_k = \left\lceil \frac{2 \log(12)}{\mu \eta_k} \right\rceil \quad \forall k \geq 1.$$

Then the time horizon $T = T_0 + \cdots + T_{K-1}$ satisfies

$$T \lesssim \frac{L}{\mu} \log \left( \frac{LD}{\sigma^2} \right)^+ + \frac{\sigma^2}{\mu \mathcal{G}} \leq \frac{L}{\mu} \log \left( \frac{D}{\mathcal{G}} \right)^+ + \frac{\sigma^2}{\mu \mathcal{G}}$$

and the corresponding tracking error satisfies $\mathbb{E}[\varphi_T(X_K) - \varphi_T^\star] \lesssim \mathcal{G}$.

## 5.4 Tracking the minimal value: high-probability guarantees

In this section, we derive the high-probability analogues of the results in Section 5.3. In light of Proposition 5.11, we seek upper bounds on the sums

$$\sum_{i=0}^{t-1} \langle z_i, x_i - x_t^\star \rangle (1 - \hat\rho)^{t-1-i}, \quad \sum_{i=0}^{t-1} \|z_i\|^2 (1 - \hat\rho)^{t-1-i}, \quad \sum_{i=0}^{t-1} G_{i,t}^2 (1 - \hat\rho)^{t-1-i}$$

that hold with high probability. The last two sums can easily be estimated under boundedness or light-tail assumptions on $\|z_i\|$ and $G_{i,t}$. Controlling the first sum is more challenging because the error $\|x_i - x_t^\star\|$ may in principle grow large. In order to control this term, we use a remarkable generalization of Freedman's inequality, recently proved in [14] for the purpose of analyzing the stochastic gradient method on static nonsmooth problems (without a regularizer).

The main idea is as follows. Fix a horizon $t$, assume $\mathbb{E}[z_i \mid \mathcal{F}_{i,t}] = 0$ for all $0 \leq i < t$ (recall that $\mathcal{F}_{i,t} := \sigma(\mathcal{F}_i, x_t^\star)$), and define the martingale difference sequence

$$d_i := \langle z_i, x_i - x_t^\star \rangle (1 - \hat\rho)^{t-1-i}$$

adapted to the filtration $(\mathcal{F}_{i+1,t})_{i=0}^{t-1}$. Roughly speaking, under mild light-tail assumptions, the total conditional variance of the corresponding martingale $\sum_{i=0}^{t-1} d_i$ can be bounded above with high probability by an affine transformation of itself, i.e., by an affine combination of the sequence $\{d_i\}_{i=0}^{t-1}$. In this way, the martingale is self-regulating. This is the content of the following proposition. The proof follows from Lemma 5.10 and algebraic manipulation and is placed in the appendix (see Section A.4).

**Proposition 5.14** (Self-regulation). *The iterates $\{x_t\}$ produced by Algorithm 1 with $r_t \equiv r$ and constant step size $\eta \leq 1/2L$ satisfy the following bound for all $\lambda \in (0, \mu\eta)$:*

$$\sum_{i=0}^{t-1} \|x_i - x_t^\star\|^2 (1 - \lambda)^{2(t-1-i)} \leq \sum_{j=0}^{t-2} \left( 2\eta \sum_{i=j+1}^{t-1} (1 - \lambda)^{t-2-i} \right) \langle z_j, x_j - x_t^\star \rangle (1 - \lambda)^{t-1-j}$$

$$+ \tfrac{1}{\lambda} (1 - \lambda)^{t-1} \|x_0 - x_t^\star\|^2 + \tfrac{2\eta^2}{\lambda} \sum_{j=0}^{t-2} \|z_j\|^2 (1 - \lambda)^{t-2-j}$$

$$+ \tfrac{\eta}{\mu\lambda} \sum_{j=0}^{t-2} G_{j,t}^2 (1 - \lambda)^{t-2-j}.$$

In order to bound the self-regulating martingale $\sum_{i=0}^{t-1} d_i$, we use the following direct consequence of the generalized Freedman inequality developed in [14].

**Theorem 5.15** (Consequence of generalized Freedman). *Let $(D_i)_{i=0}^n$ and $(V_i)_{i=0}^n$ be scalar stochastic processes on a probability space with filtration $(\mathcal{H}_i)_{i=0}^{n+1}$ satisfying*

$$\mathbb{E}[\exp(\lambda D_i) \mid \mathcal{H}_i] \le \exp(\lambda^2 V_i/2) \quad \text{for all} \quad \lambda \ge 0.$$

*Suppose that $D_i$ is $\mathcal{H}_{i+1}$-measurable with $\mathbb{E}|D_i| < \infty$ and $\mathbb{E}[D_i \mid \mathcal{H}_i] = 0$, and that $V_i$ is nonnegative and $\mathcal{H}_i$-measurable. Suppose moreover that there are constants $\alpha_0, \ldots, \alpha_n \ge 0$, $\delta \in [0,1]$, and $\beta(\delta) \ge 0$ satisfying*

$$\mathbb{P}\left\{ \sum_{i=0}^n V_i \le \sum_{i=0}^n \alpha_i D_i + \beta(\delta) \right\} \ge 1 - \delta.$$

*Set $\alpha := \max\{\alpha_0, \ldots, \alpha_n\}$. Then for all $\tau > 0$, the following bound holds:*

$$\mathbb{P}\left\{ \sum_{i=0}^n D_i \ge \tau \right\} \le \delta + \exp\left( -\frac{\tau}{4\alpha + 8\beta(\delta)/\tau} \right).$$

Combining Proposition 5.14 and Theorem 5.15 yields the following tail bound for $\sum_{i=0}^{t-1} d_i$.

**Proposition 5.16** (Noise martingale tail bound). *Suppose that Assumption 4.4 holds, let $\{x_t\}$ be the iterates produced by Algorithm 1 with constant step size $\eta \le 1/2L$, and set $\hat{\rho} := \mu\eta/(2 - \mu\eta)$. Then there is an absolute constant $c > 0$ such that for any specified $t \in \mathbb{N}$, $\delta \in (0,1)$, and $\tau > 0$, the following bound holds:*

$$\mathbb{P}\left\{ \sum_{i=0}^{t-1} \langle z_i, x_i - x_t^\star \rangle (1 - \hat{\rho})^{t-1-i} \ge \tau \right\} \le \delta + \exp\left( -\frac{\tau}{4\alpha + 8\beta_t \log(3e/\delta)/\tau} \right),$$

*where $\alpha := 3\eta(c\sigma)^2/\hat{\rho}$ and*

$$\beta_t := (1 - \hat{\rho})^{t-1} \left( \|x_0 - x_0^\star\|^2 + \Delta^2 t^2 \right) \frac{2(c\sigma)^2}{\hat{\rho}} + \frac{2\eta^2(c\sigma)^4}{\hat{\rho}^2} + \frac{3\mu\Delta^2\eta(c\sigma)^2}{\hat{\rho}^4}.$$

*Proof.* By Assumption 4.4, there exists an absolute constant $c \ge 1$ such that $\|z_i\|^2$ is sub-exponential conditioned on $\mathcal{F}_{i,t}$ with parameter $c\sigma^2$ and $z_i$ is mean-zero sub-Gaussian conditioned on $\mathcal{F}_{i,t}$ with parameter $c\sigma$ for all indices $0 \le i < t$. Then for each $0 \le i < t$, the $\mathcal{F}_{i+1,t}$-measurable random variable $\langle z_i, x_i - x_t^\star \rangle$ is mean-zero sub-Gaussian conditioned on $\mathcal{F}_{i,t}$ with parameter $c\sigma\|x_i - x_t^\star\|$, so

$$\mathbb{E}\left[ \exp\left( \lambda\langle z_i, x_i - x_t^\star \rangle(1-\hat{\rho})^{t-1-i} \right) \mid \mathcal{F}_{i,t} \right] \le \exp\left( \lambda^2(c\sigma)^2\|x_i - x_t^\star\|^2(1-\hat{\rho})^{2(t-1-i)}/2 \right) \quad \forall \lambda \in \mathbb{R}.$$

Now fix $t \ge 1$ and observe that Proposition 5.14 yields the total conditional variance bound

$$\sum_{i=0}^{t-1} (c\sigma)^2\|x_i - x_t^\star\|^2(1-\hat{\rho})^{2(t-1-i)} \le \sum_{j=0}^{t-2} \alpha_j\langle z_j, x_j - x_t^\star \rangle(1-\hat{\rho})^{t-1-j} + R_t,$$

where $0 \le \alpha_j \le \alpha$ for all $0 \le j \le t - 2$ and

$$R_t := \frac{(c\sigma)^2}{\hat{\rho}}(1-\hat{\rho})^{t-1}\|x_0 - x_t^\star\|^2 + \frac{2\eta^2(c\sigma)^2}{\hat{\rho}} \sum_{j=0}^{t-2} \|z_j\|^2(1-\hat{\rho})^{t-2-j} + \frac{\eta(c\sigma)^2}{\mu\hat{\rho}} \sum_{j=0}^{t-2} G_{j,t}^2(1-\hat{\rho})^{t-2-j}.$$

We claim that

$$\mathbb{P}\left\{ R_t \le \beta_t \log\left( \frac{3e}{\delta} \right) \right\} \ge 1 - \delta \quad \forall \delta \in (0,1). \tag{10}$$

To verify (10), observe first that for all $n \ge 0$, the sum $\sum_{i=0}^n \|z_i\|^2(1-\hat{\rho})^{n-i}$ is sub-exponential with parameter $\sum_{i=0}^n c\sigma^2(1-\hat{\rho})^{n-i} \le (c\sigma)^2/\hat{\rho}$, so Markov's inequality implies

$$\mathbb{P}\left\{ \sum_{i=0}^n \|z_i\|^2(1-\hat{\rho})^{n-i} \le \frac{(c\sigma)^2}{\hat{\rho}} \log\left( \frac{e}{\delta} \right) \right\} \ge 1 - \delta \quad \forall \delta \in (0,1). \tag{11}$$

Further, for all $0 \le n < t$, it follows from Assumption 4.4 and Lemma 2.3 that $\|x_0 - x_t^\star\|^2$ is sub-exponential with parameter $2\left( \|x_0 - x_0^\star\|^2 + \Delta^2 t^2 \right)$ and $\sum_{i=0}^n G_{i,t}^2(1-\hat{\rho})^{n-i}$ is sub-exponential

with parameter

$$\sum_{i=0}^{n}(\mu\Delta)^2(t-i)^2(1-\hat{\rho})^{n-i} = (\mu\Delta)^2(1-\hat{\rho})^{n+1-t}\sum_{i=0}^{n}(t-i)^2(1-\hat{\rho})^{t-i-1} \leq \frac{2(\mu\Delta)^2}{\hat{\rho}^3(1-\hat{\rho})^{t-1-n}},$$

so Markov's inequality implies

$$\mathbb{P}\left\{\|x_0 - x_t^\star\|^2 \leq 2\left(\|x_0 - x_0^\star\|^2 + \Delta^2 t^2\right)\log\left(\frac{e}{\delta}\right)\right\} \geq 1-\delta \quad \forall\delta \in (0,1) \tag{12}$$

and

$$\mathbb{P}\left\{\sum_{i=0}^{n} G_{i,t}^2(1-\hat{\rho})^{n-i} \leq \frac{2(\mu\Delta)^2}{\hat{\rho}^3(1-\hat{\rho})^{t-1-n}}\log\left(\frac{e}{\delta}\right)\right\} \geq 1-\delta \quad \forall\delta \in (0,1). \tag{13}$$

Thus, (11)–(13) and a union bound yield (10). Consequently, Theorem 5.15 implies that the following bound holds for all $\delta \in (0,1)$ and $\tau > 0$:

$$\mathbb{P}\left\{\sum_{i=0}^{t-1}\langle z_i, x_i - x_t^\star\rangle(1-\hat{\rho})^{t-1-i} \geq \tau\right\} \leq \delta + \exp\left(-\frac{\tau}{4\alpha + 8\beta_t\log(3e/\delta)/\tau}\right),$$

as claimed. $\qquad\square$

We may now deduce the following precise version of Theorem 4.6 using the tail bound furnished by Proposition 5.16.

**Theorem 5.17** (Function gap with high probability). *Suppose that Assumption 4.4 holds, let $\{\hat{x}_t\}$ be the iterates produced by Algorithm 2 with constant step size $\eta \leq 1/2L$, and set $\hat{\rho} := \mu\eta/(2-\mu\eta)$. Then there is an absolute constant $c > 0$ such that for any specified $t \in \mathbb{N}$ and $\delta \in (0,1)$, the following estimate holds with probability at least $1-\delta$:*

$$\varphi_t(\hat{x}_t) - \varphi_t^\star \leq (1-\hat{\rho})^t\left(\varphi_t(x_0) - \varphi_t^\star + \tfrac{\mu}{4}\|x_0 - x_t^\star\|^2\right) + \left(\eta(c\sigma)^2 + \frac{8\Delta^2}{\mu\eta^2} + 9\hat{\rho}\sqrt{8\beta_t}\right)\log\left(\frac{4e}{\delta}\right),$$

*where*

$$\beta_t := (1-\hat{\rho})^{t-1}\left(\|x_0 - x_0^\star\|^2 + \Delta^2 t^2\right)\frac{2(c\sigma)^2}{\hat{\rho}} + \frac{2\eta^2(c\sigma)^4}{\hat{\rho}^2} + \frac{3\mu\Delta^2\eta(c\sigma)^2}{\hat{\rho}^4}.$$

*Proof.* A quick computation shows that given any $\delta \in (0,1)$, we may take

$$\tau = (2+\sqrt{5})\sqrt{8\beta_t}\log\left(\frac{3e}{\delta}\right)$$

in Proposition 5.16 to obtain

$$\mathbb{P}\left\{\sum_{i=0}^{t-1}\langle z_i, x_i - x_t^\star\rangle(1-\hat{\rho})^{t-1-i} < (2+\sqrt{5})\sqrt{8\beta_t}\log\left(\frac{3e}{\delta}\right)\right\} \geq 1-2\delta. \tag{14}$$

We may now combine (11), (13), and (14) together with Proposition 5.11 and a union bound to conclude that for all $\delta \in (0,1)$, the estimate

$$\varphi_t(\hat{x}_t) - \varphi_t^\star \leq (1-\hat{\rho})^t\left(\varphi_t(x_0) - \varphi_t^\star + \tfrac{\mu}{4}\|x_0 - x_t^\star\|^2\right) + \left(\eta(c\sigma)^2 + \frac{2\mu\Delta^2}{\hat{\rho}^2}\right)\log\left(\frac{e}{\delta}\right)$$

$$+ (2+\sqrt{5})\hat{\rho}\sqrt{8\beta_t}\log\left(\frac{3e}{\delta}\right)$$

holds with probability at least $1-4\delta$; hence

$$\varphi_t(\hat{x}_t) - \varphi_t^\star \leq (1-\hat{\rho})^t\left(\varphi_t(x_0) - \varphi_t^\star + \tfrac{\mu}{4}\|x_0 - x_t^\star\|^2\right) + \left(\eta(c\sigma)^2 + \frac{8\Delta^2}{\mu\eta^2} + 9\hat{\rho}\sqrt{8\beta_t}\right)\log\left(\frac{e}{\delta}\right).$$

with probability at least $1-4\delta$. $\qquad\square$

**Remark 5.18.** To see that Theorem 5.17 entails Theorem 4.6, fix $t \in \mathbb{N}$ and observe that upon setting $C := \max\{c, 1\}$, we have

$$\hat{\rho}\sqrt{8\beta_t} \leq 4C^2 \left( \sqrt{(1-\hat{\rho})^t \left( \|x_0 - x_0^\star\|^2 + \Delta^2 t^2 \right) \mu\eta\sigma^2} + \eta\sigma^2 + \sqrt{6}\frac{\Delta\sigma}{\sqrt{\mu\eta}} \right),$$

while the AM-GM inequality implies

$$2\sqrt{(1-\hat{\rho})^t \left( \|x_0 - x_0^\star\|^2 + \Delta^2 t^2 \right) \mu\eta\sigma^2} \leq (1-\hat{\rho})^t \left( \mu\|x_0 - x_0^\star\|^2 + \mu\Delta^2 t^2 \right) + \eta\sigma^2,$$

inequality (9) implies

$$(1-\hat{\rho})^t \left( \mu\|x_0 - x_0^\star\|^2 + \mu\Delta^2 t^2 \right) \leq 2(1-\hat{\rho})^t \left( \varphi_0(x_0) - \varphi_0^\star \right) + \frac{16\Delta^2}{\mu\eta^2},$$

and Young's inequality implies

$$\frac{2\Delta\sigma}{\sqrt{\mu\eta}} \leq \eta\sigma^2 + \frac{\Delta^2}{\mu\eta^2}.$$

Hence

$$\hat{\rho}\sqrt{8\beta_t} \lesssim (1-\hat{\rho})^t \left( \varphi_0(x_0) - \varphi_0^\star \right) + \eta\sigma^2 + \frac{\Delta^2}{\mu\eta^2}.$$

Further, inequalities (7) and (9) together with Assumption 4.4 imply that for all $\delta \in (0, 1)$, the estimate

$$(1-\hat{\rho})^t \left( \varphi_t(x_0) - \varphi_t^\star + \tfrac{\mu}{4}\|x_0 - x_t^\star\|^2 \right) \leq 3(1-\hat{\rho})^t \left( \varphi_0(x_0) - \varphi_0^\star \right) + \frac{32\Delta^2}{\mu\eta^2} \log\left( \frac{e}{\delta} \right)$$

holds with probability at least $1 - \delta$. Thus, under the assumptions of Theorem 5.17, a union bound reveals that for all $t \in \mathbb{N}$ and $\delta \in (0, 1)$, the estimate

$$\varphi_t(\hat{x}_t) - \varphi_t^\star \leq (1-\hat{\rho})^t \left( \varphi_t(x_0) - \varphi_t^\star + \tfrac{\mu}{4}\|x_0 - x_t^\star\|^2 \right) + \left( \eta(c\sigma)^2 + \frac{8\Delta^2}{\mu\eta^2} + 9\hat{\rho}\sqrt{8\beta_t} \right) \log\left( \frac{8e}{\delta} \right)$$

$$\lesssim \left( (1-\hat{\rho})^t \left( \varphi_0(x_0) - \varphi_0^\star \right) + \eta\sigma^2 + \frac{\Delta^2}{\mu\eta^2} \right) \log\left( \frac{e}{\delta} \right)$$

holds with probability at least $1 - \delta$.

We may now apply Theorem 4.6 to obtain the formal version of Theorem 4.7; the proof is analogous to that of Theorem 5.8 and appears in the appendix (see Section A.5).

**Theorem 5.19** (Time to track with high probability). *Suppose that Assumption 4.4 holds and that we are in the low drift-to-noise regime $\Delta/\sigma < \sqrt{\mu/16L^3}$. Set $\eta_\star = (2\Delta^2/\mu\sigma^2)^{1/3}$ and $\mathcal{G} = \mu(\Delta\sigma^2/\mu^2)^{2/3}$. Suppose moreover that we have available a positive upper bound on the initial gap $D \geq \varphi_0(x_0) - \varphi_0^\star$. Fix $\delta \in (0, 1)$ and consider running Algorithm 2 in $k = 0, \ldots, K-1$ epochs, namely, set $X_0 = x_0$ and iterate the process*

$$X_{k+1} = \overline{\mathrm{PSG}}(X_k, \mu, \eta_k, T_k) \quad \text{for} \quad k = 0, \ldots, K-1,$$

*where the number of epochs is*

$$K = 1 + \left\lceil \log_2 \left( \frac{1}{L} \cdot \left( \frac{\sigma^2\mu}{\Delta^2} \right)^{1/3} \right) \right\rceil$$

*and we set*

$$\eta_0 = \frac{1}{2L}, \quad T_0 = \left\lceil \frac{4L}{\mu} \log\left( \frac{LD}{\sigma^2} \right)^+ \right\rceil \quad \text{and} \quad \eta_k = \frac{\eta_{k-1} + \eta_\star}{2}, \quad T_k = \left\lceil \frac{2 \log\left( 4c \log(e/\delta) \right)^+}{\mu\eta_k} \right\rceil$$

*for all $k \geq 1$, where $c > 0$ is the absolute constant furnished by the bound (3). Then the time horizon $T = T_0 + \cdots + T_{K-1}$ satisfies*

$$T \lesssim \frac{L}{\mu} \log\left( \frac{LD}{\sigma^2} \right)^+ + \frac{\sigma^2}{\mu\mathcal{G}} \left( 1 \vee \log\log\frac{e}{\delta} \right) \leq \frac{L}{\mu} \log\left( \frac{D}{\mathcal{G}} \right)^+ + \frac{\sigma^2}{\mu\mathcal{G}} \left( 1 \vee \log\log\frac{e}{\delta} \right)$$

*and the corresponding tracking error satisfies*

$$\varphi_T(X_K) - \varphi_T^\star \lesssim \mathcal{G} \log\left(\frac{e}{\delta}\right)$$

*with probability at least* $1 - K\delta$.

## 6  Numerical illustrations

We investigate the empirical behavior of our finite-time bounds on numerical examples with synthetic data. We consider examples of a) least-squares recovery; b) sparse least-squares recovery; c) $\ell_2^2$-regularized logistic regression; and investigate the behavior of $\|x_t - x_t^\star\|^2$ and $\varphi_t(\hat{x}_t) - \varphi_t^\star$ in each case. The main findings are that our bounds exhibit: 1) the correct dependence on $\eta$, $\sigma$, and $\Delta$; 2) excellent coverage in Monte-Carlo simulations. Code is available online at https://github.com/joshuacutler/TimeDriftExperiments.

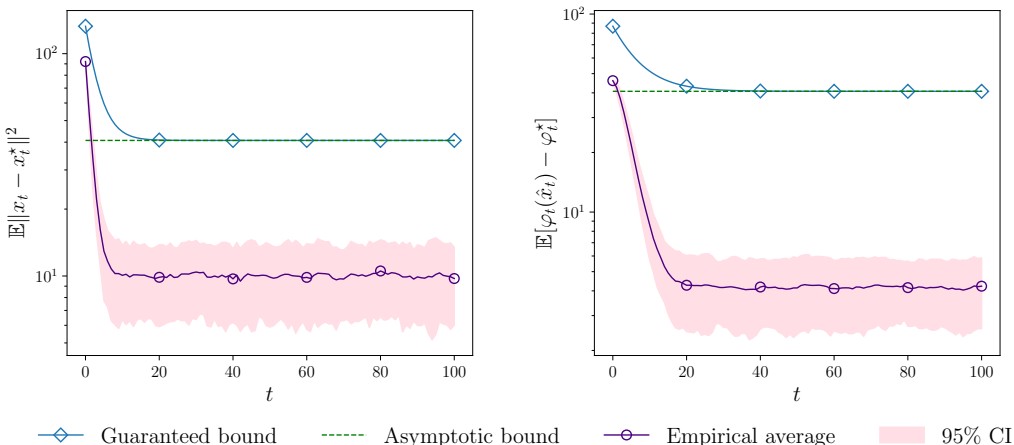

Figure 1: Semilog plots of guaranteed bounds and empirical tracking errors with respect to iteration $t$ for least-squares recovery. Shaded regions indicate the 95% confidence intervals for $\|x_t - x_t^\star\|^2$ and $\varphi_t(\hat{x}_t) - \varphi_t^\star$; empirical averages and confidence intervals are computed over 100 trials. Default parameter values: $\mu = 1$, $L = 1$, $\sigma = 10$, $\Delta = 1$, and $\eta = \eta_\star$.

**Least-squares recovery.**  Fix $x_0, x_0^\star \in \mathbb{R}^d$ with standard Gaussian entries, and consider a Gaussian random walk $(x_t^\star)$ given by $x_{t+1}^\star = x_t^\star + v_t$, where $v_t$ is drawn uniformly from the sphere of radius $\Delta$ in $\mathbb{R}^d$. Given a fixed matrix $A \in \mathbb{R}^{n \times d}$, we aim to recover the vectors $(x_t^\star)$ via least-squares:

$$\min_{x \in \mathbb{R}^d} \mathbb{E}_{w \sim \mathcal{P}_t} \frac{1}{2}\|Ax - w\|^2,$$

where $\mathcal{P}_t = \mathcal{N}(Ax_t^\star, C)$ with $C = (\sigma^2/n\|A\|_{\mathrm{op}}^2)I_n$. This amounts to the target problem (1) under the identifications $f_t(x) = \mathbb{E}_{w \sim \mathcal{P}_t} \frac{1}{2}\|Ax - w\|^2$ and $r_t = 0$; clearly $\|x_t^\star - x_{t+1}^\star\| = \Delta$ and $\sup_x \|\nabla f_t(x) - \nabla f_{t+1}(x)\| \leq \|A\|_{\mathrm{op}}^2\Delta$. We implement Algorithms 1 and 2 initialized at $x_0$ using the sample gradient $g_t = A^T(Ax_t - w)$ at step $t$, where $w \sim \mathcal{P}_t$; hence $\mathbb{E}\|\nabla f_t(x_t) - g_t\|^2 \leq \sigma^2$.

In our simulations, we take $d = 50$, $n = 100$, and randomly generate $A$ via its singular value decomposition (using Haar-distributed orthogonal matrices) so that its minimal singular value is $\sqrt{\mu}$ and its maximal singular value is $\sqrt{L}$. In Figures 1 and 2, we use default parameter values $\mu = 1$, $L = 1$, $\sigma = 10$, $\Delta = 1$, and the corresponding asymptotically optimal step size $\eta = \eta_\star$. Since $f_t$ is $\mu$-strongly convex and $L$-smooth, this puts us in the low drift/noise regime in Figure 1: $\Delta/\sigma < \sqrt{\mu/16L^3} = 1/4$. To estimate the empirical averages and confidence intervals of $\|x_t - x_t^\star\|^2$ and $\varphi_t(\hat{x}_t) - \varphi_t^\star$, we run 100 trials with horizon $T = 100$. The results confirm our bounds and show that they capture the correct dependence on $\eta$, $\sigma$, and $\Delta$.

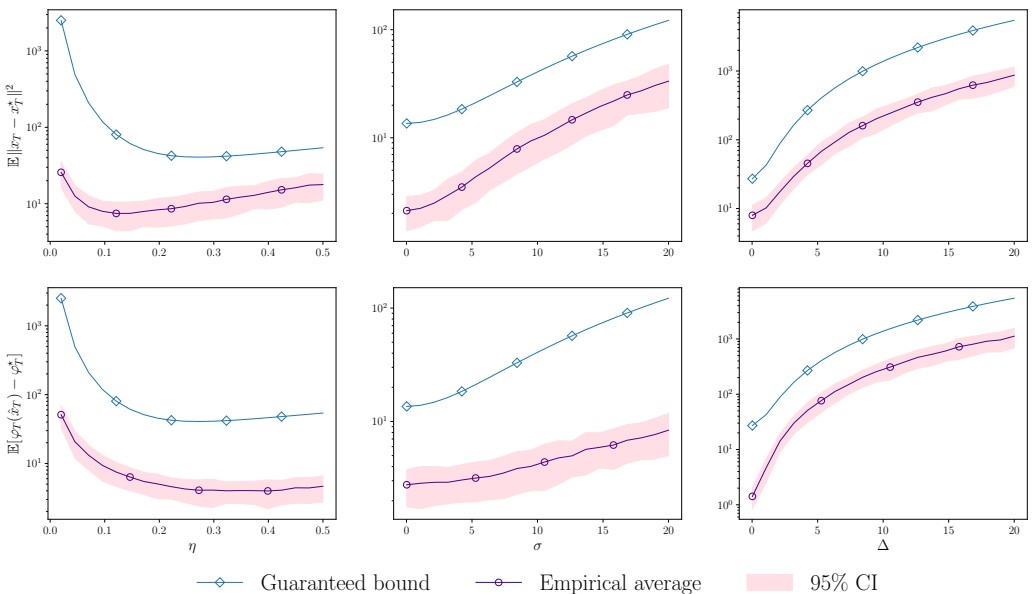

Figure 2: Semilog plots of guaranteed bounds and empirical tracking errors at horizon $T = 100$ with respect to $\eta$, $\sigma$, and $\Delta$ for least-squares recovery. Shaded regions indicate the 95% confidence intervals for $\|x_T - x_T^\star\|^2$ and $\varphi_T(\hat{x}_T) - \varphi_T^\star$; empirical averages and confidence intervals are computed over 100 trials. Default parameter values: $\mu = 1$, $L = 1$, $\sigma = 10$, $\Delta = 1$, and $\eta = \eta_\star$.

**Sparse least-squares recovery.** Next, we consider least-squares recovery constrained to the closed $\ell_1$-ball in $\mathbb{R}^d$, which we denote by $B_1$. We aim to recover a sparse sequence of vectors in $B_1$ defined as follows. Set $s = \lfloor \log d \rfloor$, draw a vector $u$ uniformly from the $\ell_1$-ball in $\mathbb{R}^s$, fix $x_0^\star = (u, 0) \in \mathbb{R}^d$, and select $\Delta \in (0, \sqrt{2}]$. At step $t$, with probability $p = (4 - 2\Delta^2)/(4 - \Delta^2)$, we set $x_{t+1}^\star = x_t^\star + v$, where $v$ is selected to have the same support as $x_t^\star$ and satisfy $\|v\| = \Delta/\sqrt{2}$ and $x_t^\star + v \in B_1$; otherwise, with probability $1 - p$, we obtain $x_{t+1}^\star$ from $x_t^\star$ by swapping precisely one nonzero coordinate with a zero coordinate. Then the resulting sparse sequence $(x_t^\star)$ in $B_1$ satisfies $\mathbb{E}\|x_t^\star - x_{t+1}^\star\|^2 \leq \Delta^2$. Given a fixed matrix $A \in \mathbb{R}^{n \times d}$, we aim to recover $(x_t^\star)$ via constrained least-squares:

$$\min_{x \in B_1} \mathbb{E}_{w \sim \mathcal{P}_t} \tfrac{1}{2}\|Ax - w\|^2,$$

where $\mathcal{P}_t = \mathcal{N}\left(Ax_t^\star, C\right)$ with $C = (\sigma^2/n\|A\|_{\mathrm{op}}^2)I_n$. This amounts to the target problem (1) under the identifications $f_t(x) = \mathbb{E}_{w \sim \mathcal{P}_t} \tfrac{1}{2}\|Ax - w\|^2$ and $r_t = \delta_{B_1}$ (the convex indicator of $B_1$); clearly $\mathbb{E}[\sup_x \|\nabla f_t(x) - \nabla f_{t+1}(x)\|^2] \leq (\|A\|_{\mathrm{op}}^2 \Delta)^2$. Fixing $x_0$ drawn uniformly from $B_1$, we implement Algorithms 1 and 2 initialized at $x_0$ using the sample gradient $g_t = A^T(Ax_t - w)$ at step $t$, where $w \sim \mathcal{P}_t$; hence $\mathbb{E}\|\nabla f_t(x_t) - g_t\|^2 \leq \sigma^2$.

In our simulations, we take $d = 50$, $n = 100$, and randomly generate $A$ via its singular value decomposition (using Haar-distributed orthogonal matrices) so that its minimal singular value is $\sqrt{\mu}$ and its maximal singular value is $\sqrt{L}$. In Figures 3 and 4, we use default parameter values $\mu = 1$, $L = 1$, $\sigma = 1/2$, $\Delta = 1/20$, and the corresponding asymptotically optimal step size $\eta = \eta_\star$. Since $f_t$ is $\mu$-strongly convex and $L$-smooth, this puts us in the low drift/noise regime in Figure 3: $\Delta/\sigma < \sqrt{\mu/16L^3} = 1/4$. To estimate the empirical averages and confidence intervals of $\|x_t - x_t^\star\|^2$ and $\varphi_t(\hat{x}_t) - \varphi_t^\star$, we run 100 trials with horizon $T = 100$. The results confirm our bounds and show that they capture the correct dependence on $\eta$, $\sigma$, and $\Delta$.

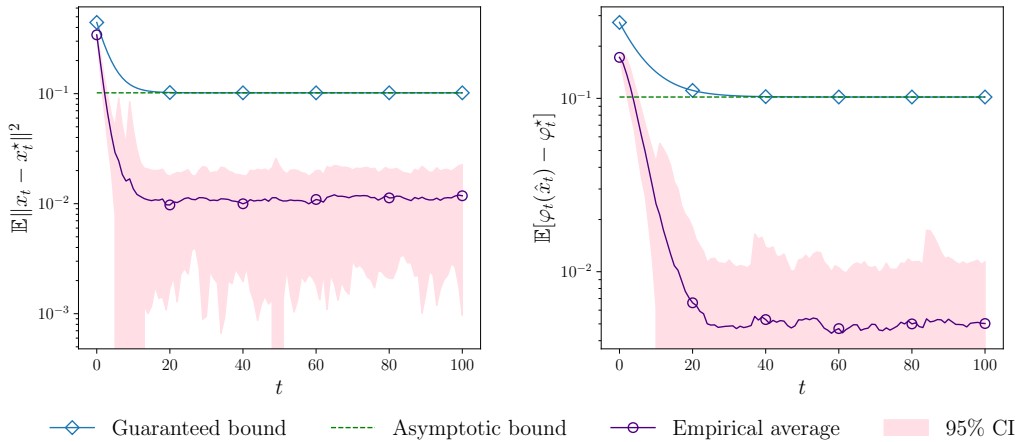

Figure 3: Semilog plots of guaranteed bounds and empirical tracking errors with respect to iteration $t$ for sparse least-squares recovery. Shaded regions indicate the $95\%$ confidence intervals for $\|x_t - x_t^\star\|^2$ and $\varphi_t(\hat{x}_t) - \varphi_t^\star$; empirical averages and confidence intervals are computed over 100 trials. Default parameter values: $\mu = 1$, $L = 1$, $\sigma = 1/2$, $\Delta = 1/20$, and $\eta = \eta_\star$.

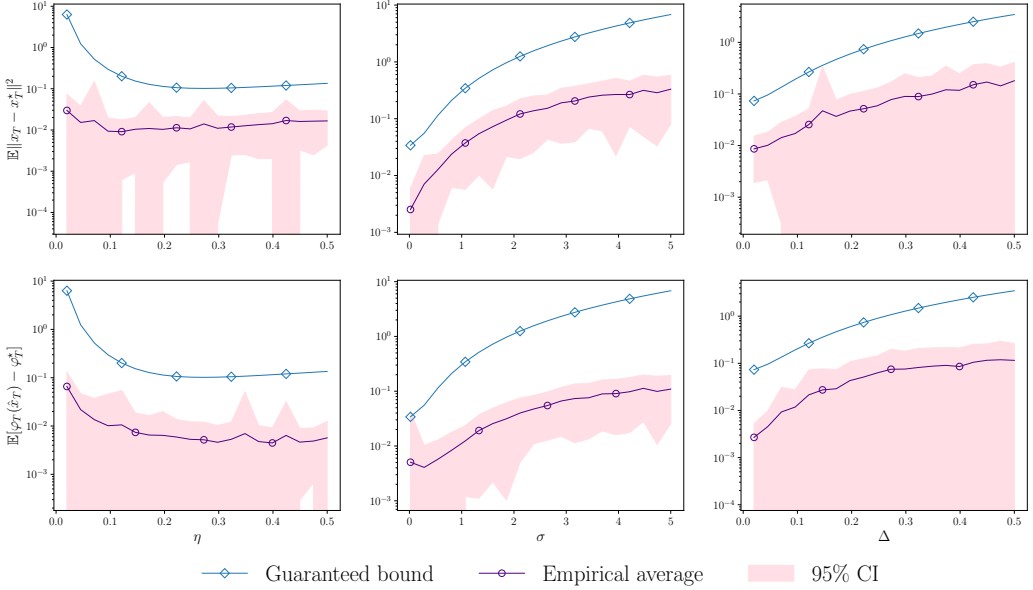

Figure 4: Semilog plots of guaranteed bounds and empirical tracking errors at horizon $T = 100$ with respect to $\eta$, $\sigma$, and $\Delta$ for sparse least-squares recovery. Shaded regions indicate the $95\%$ confidence intervals for $\|x_T - x_T^\star\|^2$ and $\varphi_T(\hat{x}_T) - \varphi_T^\star$; empirical averages and confidence intervals are computed over 100 trials. Default parameter values: $\mu = 1$, $L = 1$, $\sigma = 1/2$, $\Delta = 1/20$, and $\eta = \eta_\star$.

$\ell_2^2$**-regularized logistic regression.**    Finally, we consider the time-varying $\ell_2^2$-regularized logistic regression problem

$$\min_{x \in \mathbb{R}^d} \frac{1}{n} \left( \sum_{i=1}^n \log(1 + \exp\langle a_i, x \rangle) - \langle Ax, b_t \rangle \right) + \frac{\mu}{2} \|x\|^2,$$

where the fixed matrix $A \in \mathbb{R}^{n \times d}$ has standard Gaussian rows $a_1, ..., a_n \in \mathbb{R}^d$, $(b_t)$ is a random sequence of label vectors in $\{0, 1\}^n$ such that $b_t$ and $b_{t+1}$ differ in precisely one coordinate for each $t$, and $\mu > 0$. This amounts to the target problem (1) under the identifications $f_t(x) = \frac{1}{n} \left( \sum_{i=1}^n \log(1 + \exp\langle a_i, x \rangle) - \langle Ax, b_t \rangle \right) + \frac{\mu}{2} \|x\|^2$ and $r_t = 0$; setting $L = \frac{1}{4n} \|A\|_{\text{op}}^2 + \mu$, it follows

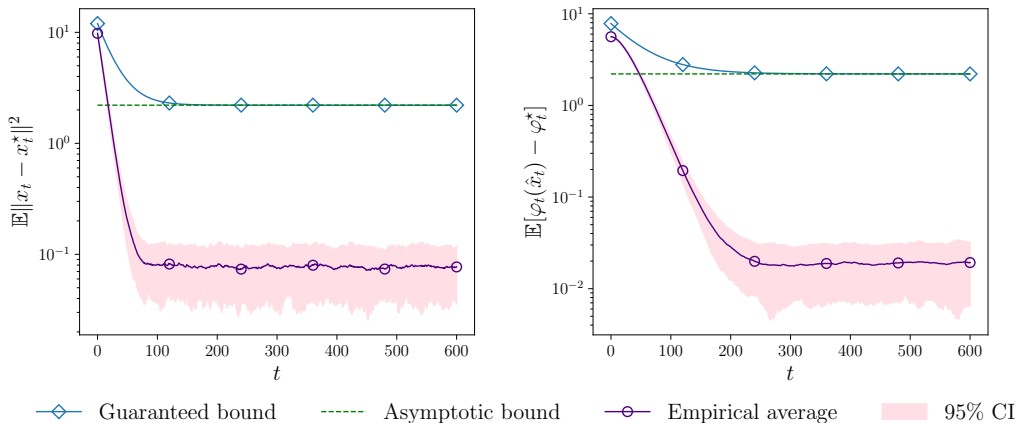

Figure 5: Semilog plots of guaranteed bounds and empirical tracking errors with respect to iteration $t$ for $\ell_2^2$-regularized logistic regression. Shaded regions indicate the 95% confidence intervals for $\|x_t - x_t^\star\|^2$ and $\varphi_t(\hat{x}_t) - \varphi_t^\star$; empirical averages and confidence intervals are computed over 100 trials. Default parameter values: $\mu = 1$ and $\eta = \eta_\star$.

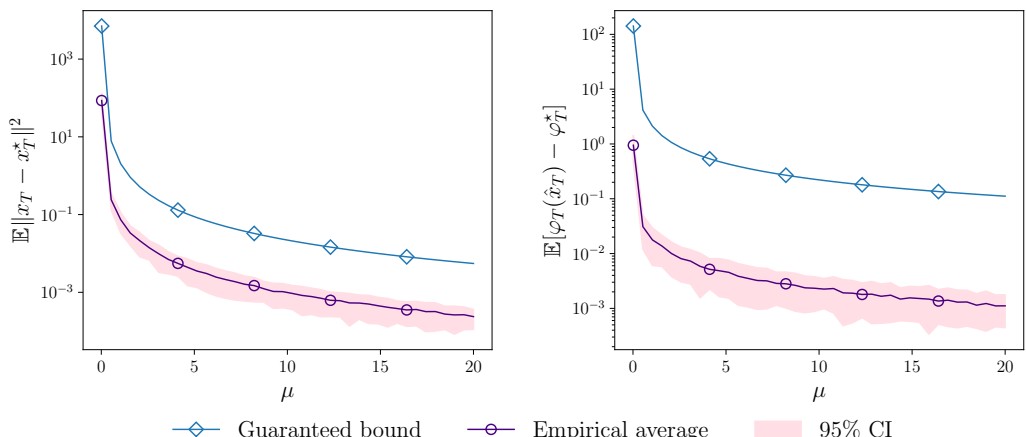

Figure 6: Semilog plots of guaranteed bounds and empirical tracking errors at horizon $T = 600$ with respect to the strong convexity parameter $\mu$ for $\ell_2^2$-regularized logistic regression. Shaded regions indicate the 95% confidence intervals for $\|x_T - x_T^\star\|^2$ and $\varphi_T(\hat{x}_T) - \varphi_T^\star$; empirical averages and confidence intervals are computed over 100 trials, using the asymptotically optimal step size $\eta_\star$ (which itself depends on $\mu$).

that $f_t$ is $\mu$-strongly convex and $L$-smooth. Letting $(x_t^\star)$ denote the corresponding sequence of minimizers and setting $\Delta = \frac{1}{\mu n} \max_{i=1,\dots,n} \|a_i\|$, it follows that $\sup_x \|\nabla f_t(x) - \nabla f_{t+1}(x)\| \leq \mu \Delta$ and hence $\|x_t^\star - x_{t+1}^\star\| \leq \Delta$. Fixing the initial label $b_0$ (drawn uniformly from $\{0, 1\}^n$) and a standard Gaussian vector $x_0 \in \mathbb{R}^d$, we implement Algorithms 1 and 2 initialized at $x_0$ using the following random summand sample gradient at each step $t$:

$$g_t = \left( \frac{\exp\langle a_k, x_t \rangle}{1 + \exp\langle a_k, x_t \rangle} - b_t^k \right) a_k + \mu x_t,$$

where $k \sim \mathcal{U}\{1, \dots, n\}$ and $b_t^k$ denotes the $k^{\text{th}}$ coordinate of $b_t$. Then $\mathbb{E}\|\nabla f_t(x_t) - g_t\|^2 \leq \sigma^2$, where

$$\sigma^2 = \frac{1}{n^2} \left( (n-2) \sum_{i=1}^n \|a_i\|^2 + \sum_{i,j=1}^n \|a_i\| \|a_j\| \right) \leq \max_{i=1,\dots,n} 2\|a_i\|^2.$$

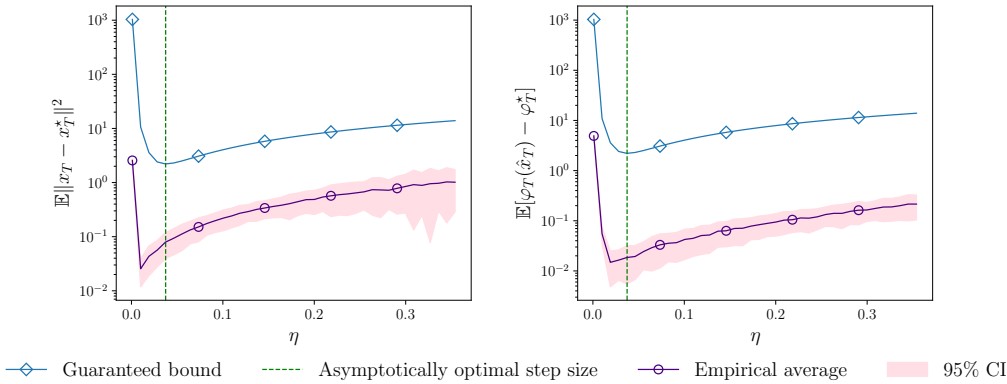

Figure 7: Semilog plots of guaranteed bounds and empirical tracking errors at horizon $T = 600$ with respect to the step size $\eta$ for $\ell_2^2$-regularized logistic regression. Shaded regions indicate the 95% confidence intervals for $\|x_T - x_T^\star\|^2$ and $\varphi_T(\hat{x}_T) - \varphi_T^\star$; empirical averages and confidence intervals are computed over 100 trials. Default parameter value: $\mu = 1$. Observe that $\eta_\star$ is close to empirically optimal.

In our simulations, we take $d = 20$ and $n = 200$, and we generate $b_{t+1}$ from $b_t$ by flipping a single coordinate selected uniformly at random. In Figure 5, we use default parameter values $\mu = 1$ and the corresponding asymptotically optimal step size $\eta = \eta_\star$. In Figure 6, we illustrate the dependence of tracking error on the regularization parameter $\mu$; here, the asymptotically optimal step size $\eta_\star$ is used (which itself depends on $\mu$). In Figure 7, we use the default parameter value $\mu = 1$. To estimate the empirical averages and confidence intervals of $\|x_t - x_t^\star\|^2$ and $\varphi_t(\hat{x}_t) - \varphi_t^\star$, we run 100 trials with horizon $T = 600$. The results confirm our bounds and show that they capture the correct dependence on $\mu$ and $\eta$. In particular, Figure 7 illustrates that $\eta_\star$ is close to empirically optimal.

## Acknowledgments

This work was supported by NSF CCF-2019844, NSF DMS-2023166, NSF DMS-1651851, CIFAR-LMB, and faculty research awards. Part of this work was done while Z. Harchaoui was visiting the Simons Institute for the Theory of Computing. Added Research of Drusvyatskiy was supported by the NSF DMS 1651851 and CCF 2023166 awards.

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

# A Additional proofs

## A.1 Proof of Theorem 5.8

For each index $k$, let $t_k := T_0 + \cdots + T_{k-1}$ (with $t_0 := 0$), $X_k^\star$ be the minimizer of the corresponding function $\varphi_{t_k}$, and

$$E_k := c\left(\frac{\eta_k \sigma^2}{\mu} + \left(\frac{\Delta}{\mu \eta_\star}\right)^2\right),$$

where $c \geq 1$ is an absolute constant satisfying the bound (2) in Theorem 3.7. Taking into account $\eta_k \geq \eta_\star$ and our selection of $c$, Theorem 3.7 implies that for any specified $k \geq 0$ and $\delta \in (0,1)$ the following estimate holds with probability at least $1 - \delta$:

$$\|X_{k+1} - X_{k+1}^\star\|^2 \leq \left(1 - \frac{\mu \eta_k}{2}\right)^{T_k} \|X_k - X_k^\star\|^2 + c\left(\frac{\eta_k \sigma^2}{\mu} + \left(\frac{\Delta}{\mu \eta_k}\right)^2\right) \log\left(\frac{e}{\delta}\right)$$

$$\leq e^{-\mu \eta_k T_k/2}\|X_k - X_k^\star\|^2 + E_k \log\left(\frac{e}{\delta}\right).$$

We will verify by induction that for all indices $k \geq 1$, the estimate $\|X_k - X_k^\star\|^2 \leq 3E_{k-1} \log(e/\delta)$ holds with probability at least $1 - k\delta$ for all $\delta \in (0,1)$. To see the base case, observe that the estimate

$$\|X_1 - X_1^\star\|^2 \leq e^{-\mu \eta_0 T_0/2}\|X_0 - X_0^\star\|^2 + E_0 \log\left(\frac{e}{\delta}\right) \leq 3E_0 \log\left(\frac{e}{\delta}\right)$$

holds with probability at least $1 - \delta$ for all $\delta \in (0,1)$. Now assume the claim holds for some index $k \geq 1$, and let $\delta \in (0,1)$; then $\|X_k - X_k^\star\|^2 \leq 3E_{k-1} \log(e/\delta)$ with probability at least $1 - k\delta$. Thus, since we also have

$$\|X_{k+1} - X_{k+1}^\star\|^2 \leq e^{-\mu \eta_k T_k/2}\|X_k - X_k^\star\|^2 + E_k \log\left(\frac{e}{\delta}\right)$$

$$\leq \frac{1}{4}\|X_k - X_k^\star\|^2 + E_k \log\left(\frac{e}{\delta}\right)$$

$$\leq \frac{E_k}{2E_{k-1}}\|X_k - X_k^\star\|^2 + E_k \log\left(\frac{e}{\delta}\right)$$

with probability at least $1 - \delta$, a union bound reveals $\|X_{k+1} - X_{k+1}^\star\|^2 \leq 3E_k \log(e/\delta)$ with probability at least $1 - (k+1)\delta$, thereby completing the induction. Hence, upon fixing $\delta \in (0,1)$, we have $\|X_K - X_K^\star\|^2 \leq 3E_{K-1} \log(e/\delta)$ with probability at least $1 - K\delta$.

Next, observe

$$\frac{2}{c}E_{K-1} - \sqrt[3]{54}\left(\frac{\Delta \sigma^2}{\mu^2}\right)^{2/3} = \frac{2\sigma^2}{\mu}(\eta_{K-1} - \eta_\star) = \frac{2\sigma^2}{\mu} \cdot \frac{\eta_0 - \eta_\star}{2^{K-1}} \leq \left(\frac{\Delta \sigma^2}{\mu^2}\right)^{2/3} = \mathcal{E},$$

so

$$\|X_K - X_K^\star\|^2 \leq \frac{3c}{2}\left(1 + \sqrt[3]{54}\right)\mathcal{E} \log\left(\frac{e}{\delta}\right) \asymp \mathcal{E} \log\left(\frac{e}{\delta}\right)$$

with probability at least $1 - K\delta$. Finally, note

$$T \lesssim \frac{L}{\mu} \log\left(\frac{\mu L D}{\sigma^2}\right)^+ + \frac{1}{\mu}\sum_{k=1}^{K-1}\frac{1}{\eta_k}$$

and

$$\sum_{k=1}^{K-1}\frac{1}{\eta_k} \leq 2L\sum_{k=1}^{K-1} 2^k \leq 2L \cdot 2^K = 8L \cdot 2^{K-2} \leq 8\left(\frac{\sigma^2 \mu}{\Delta^2}\right)^{1/3} = \frac{8\sigma^2}{\mu} \cdot \left(\frac{\Delta \sigma^2}{\mu^2}\right)^{-2/3} \asymp \frac{\sigma^2}{\mu \mathcal{E}}.$$

This completes the proof.

## A.2 The averaging lemma

We will use a small variation of the averaging lemma in [11]. To this end, consider a convex function $h: \mathbb{R}^d \to \mathbb{R} \cup \{\infty\}$ and let $\{x_t\}_{t \geq 0}$ be a sequence of vectors in $\operatorname{dom} h$. Suppose that there are constants $c_1, c_2 \in \mathbb{R}$, a nonnegative sequence of weights $\{\rho_t\}_{t \geq 1}$, and scalar sequences $\{V_t\}_{t \geq 0}$ and $\{\omega_t\}_{t \geq 1}$ satisfying the recursion

$$\rho_t h(x_t) \leq (1 - c_1 \rho_t) V_{t-1} - (1 + c_2 \rho_t) V_t + \omega_t \tag{15}$$

for all $t \geq 1$. The goal is to bound the function value $h(\hat{x}_t)$ evaluated along an "average iterate" $\hat{x}_t$.

Suppose that the relations $c_1 + c_2 > 0$, $1 - c_1 \rho_t > 0$, and $1 + c_2 \rho_t > 0$ hold for all $t \geq 1$. Define the augmented weights and products

$$\hat{\rho}_t = \frac{\rho_t (c_1 + c_2)}{1 + c_2 \rho_t} \quad \text{and} \quad \hat{\Gamma}_t = \prod_{i=1}^{t} (1 - \hat{\rho}_i)$$

for each $t \geq 1$, while setting $\hat{\Gamma}_0 = 1$. A straightforward induction yields the relation

$$1 + \sum_{i=1}^{t} \frac{\hat{\rho}_i}{\hat{\Gamma}_i} = \frac{1}{\hat{\Gamma}_t}.$$

Now set $\hat{x}_0 = x_0$ and recursively define the average iterates

$$\hat{x}_t = (1 - \hat{\rho}_t) \hat{x}_{t-1} + \hat{\rho}_t x_t$$

for all $t \geq 1$. Unrolling this recursion, we may equivalently write

$$\hat{x}_t = \hat{\Gamma}_t \left( x_0 + \sum_{i=1}^{t} \frac{\hat{\rho}_i}{\hat{\Gamma}_i} x_i \right). \tag{16}$$

The following is the key estimate we will need.

**Lemma A.1** (Averaging). *The following estimate holds for all $t \geq 0$:*

$$\frac{h(\hat{x}_t)}{c_1 + c_2} + V_t \leq \hat{\Gamma}_t \left( \frac{h(x_0)}{c_1 + c_2} + V_0 + \sum_{i=1}^{t} \frac{\omega_i}{\hat{\Gamma}_i (1 + c_2 \rho_i)} \right).$$

*Proof.* Observe that (16) expresses $\hat{x}_t$ as a convex combination of $x_0, \ldots, x_t$. Therefore, by the convexity of $h$ we may apply Jensen's inequality to obtain

$$h(\hat{x}_t) \leq \hat{\Gamma}_t h(x_0) + \sum_{i=1}^{t} \frac{\hat{\Gamma}_t \hat{\rho}_i}{\hat{\Gamma}_i} h(x_i).$$

On the other hand, for each $i \geq 1$, we may divide the recursion (15) by $\hat{\Gamma}_i (1 + c_2 \rho_i)$ to obtain

$$\frac{\hat{\rho}_i}{\hat{\Gamma}_i (c_1 + c_2)} h(x_i) \leq \frac{V_{i-1}}{\hat{\Gamma}_{i-1}} - \frac{V_i}{\hat{\Gamma}_i} + \frac{\omega_i}{\hat{\Gamma}_i (1 + c_2 \rho_i)},$$

which telescopes to yield

$$\frac{1}{c_1 + c_2} \sum_{i=1}^{t} \frac{\hat{\rho}_i}{\hat{\Gamma}_i} h(x_i) \leq V_0 - \frac{V_t}{\hat{\Gamma}_t} + \sum_{i=1}^{t} \frac{\omega_i}{\hat{\Gamma}_i (1 + c_2 \rho_i)}.$$

Hence

$$\frac{h(\hat{x}_t)}{c_1 + c_2} \leq \hat{\Gamma}_t \left( \frac{h(x_0)}{c_1 + c_2} + V_0 - \frac{V_t}{\hat{\Gamma}_t} + \sum_{i=1}^{t} \frac{\omega_i}{\hat{\Gamma}_i (1 + c_2 \rho_i)} \right),$$

as claimed. □

## A.3 Proof of Theorem 5.13

For each index $k$, let $t_k := T_0 + \cdots + T_{k-1}$ (with $t_0 := 0$) and $G_k := \eta_k \sigma^2 + 8\Delta^2/\mu\eta_\star^2$. Then taking into account $\eta_k \geq \eta_\star$, Corollary 5.12 and inequality (7) directly imply

$$\mathbb{E}\big[\varphi_{t_{k+1}}(X_{k+1}) - \varphi_{t_{k+1}}^\star\big] \leq \Big(1 - \frac{\mu\eta_k}{2}\Big)^{T_k} \mathbb{E}\big[3\big(\varphi_{t_k}(X_k) - \varphi_{t_k}^\star\big) + 2\mu\Delta^2 T_k^2\big] + \eta_k\sigma^2 + \frac{8\Delta^2}{\mu\eta_k^2}$$

$$\leq 3e^{-\mu\eta_k T_k/2}\mathbb{E}\big[\varphi_{t_k}(X_k) - \varphi_{t_k}^\star\big] + 2e^{-\mu\eta_k T_k/2}\mu\Delta^2 T_k^2 + G_k.$$

We will verify by induction that the estimate $\mathbb{E}\big[\varphi_{t_{k+1}}(X_{k+1}) - \varphi_{t_{k+1}}^\star\big] \leq 7G_k$ holds for all indices $k$. To see the base case, observe that inequality (9) facilitates the estimation

$$\mathbb{E}\big[\varphi_{t_1}(X_1) - \varphi_{t_1}^\star\big] \leq 3e^{-\mu\eta_0 T_0/2}(\varphi_0(x_0) - \varphi_0^\star) + 2e^{-\mu\eta_0 T_0/2}\mu\Delta^2 T_0^2 + G_0 \leq 7G_0.$$

Assume next that the claim holds for index $k - 1$. We then conclude

$$\mathbb{E}\big[\varphi_{t_{k+1}}(X_{k+1}) - \varphi_{t_{k+1}}^\star\big] \leq 3e^{-\mu\eta_k T_k/2}\mathbb{E}\big[\varphi_{t_k}(X_k) - \varphi_{t_k}^\star\big] + 2e^{-\mu\eta_k T_k/2}\mu\Delta^2 T_k^2 + G_k$$

$$\leq \frac{1}{4}\mathbb{E}\big[\varphi_{t_k}(X_k) - \varphi_{t_k}^\star\big] + \frac{16\Delta^2}{\mu\eta_k^2} + G_k$$

$$\leq \frac{G_k}{2G_{k-1}}\mathbb{E}\big[\varphi_{t_k}(X_k) - \varphi_{t_k}^\star\big] + \frac{16\Delta^2}{\mu\eta_k^2} + G_k \leq 7G_k,$$

completing the induction. Hence $\mathbb{E}\big[\varphi_T(X_K) - \varphi_T^\star\big] \leq 7G_{K-1}$.

Next, observe

$$G_{K-1} - \sqrt[3]{250} \cdot \mu\Big(\frac{\Delta\sigma^2}{\mu^2}\Big)^{2/3} = \sigma^2(\eta_{K-1} - \eta_\star) = \sigma^2 \cdot \frac{\eta_0 - \eta_\star}{2^{K-1}} \leq \frac{\mu}{2}\Big(\frac{\Delta\sigma^2}{\mu^2}\Big)^{2/3} = \frac{1}{2}\mathcal{G},$$

so

$$\mathbb{E}\big[\varphi_T(X_K) - \varphi_T^\star\big] \leq 7\big(\tfrac{1}{2} + \sqrt[3]{250}\big) \cdot \mu\Big(\frac{\Delta\sigma^2}{\mu^2}\Big)^{2/3} \asymp \mathcal{G}.$$

Finally, note

$$T \lesssim \frac{L}{\mu}\log\Big(\frac{LD}{\sigma^2}\Big)^+ + \frac{1}{\mu}\sum_{k=1}^{K-1}\frac{1}{\eta_k}$$

and

$$\sum_{k=1}^{K-1}\frac{1}{\eta_k} \leq 2L\sum_{k=1}^{K-1}2^k \leq 2L \cdot 2^K = 8L \cdot 2^{K-2} \leq 8\Big(\frac{\sigma^2\mu}{\Delta^2}\Big)^{1/3} = 8\sigma^2 \cdot \mu^{-1}\Big(\frac{\Delta\sigma^2}{\mu^2}\Big)^{-2/3} \asymp \frac{\sigma^2}{\mathcal{G}}.$$

This completes the proof.

## A.4 Proof of Proposition 5.14

Fix $t \geq 1$. Given $i \geq 1$ and $\alpha > 0$, the $\mu$-strong convexity of $\varphi_t$ and Lemma 5.10 imply

$$\mu\eta\|x_i - x_t^\star\|^2 \leq 2\eta(\varphi_t(x_i) - \varphi_t^\star) \leq (1 - \mu\eta)\|x_{i-1} - x_t^\star\|^2 - (1 - \alpha\eta)\|x_i - x_t^\star\|^2$$

$$+ 2\eta\langle z_{i-1}, x_{i-1} - x_t^\star\rangle + 2\eta^2\|z_{i-1}\|^2 + \tfrac{\eta}{\alpha}G_{i-1,t}^2,$$

hence

$$\big(1 + (\mu - \alpha)\eta\big)\|x_i - x_t^\star\|^2 \leq (1 - \mu\eta)\|x_{i-1} - x_t^\star\|^2 + 2\eta\langle z_{i-1}, x_{i-1} - x_t^\star\rangle$$

$$+ 2\eta^2\|z_{i-1}\|^2 + \tfrac{\eta}{\alpha}G_{i-1,t}^2.$$

Taking $\alpha = \mu$, we obtain

$$\|x_i - x_t^\star\|^2 \leq (1 - \mu\eta)\|x_{i-1} - x_t^\star\|^2 + 2\eta\langle z_{i-1}, x_{i-1} - x_t^\star\rangle + 2\eta^2\|z_{i-1}\|^2 + \tfrac{\eta}{\mu}G_{i-1,t}^2.$$

Thus, given any $\lambda \in (0, \mu\eta]$ and proceeding by induction, we conclude

$$\|x_i - x_t^\star\|^2 \leq (1-\lambda)^i \|x_0 - x_t^\star\|^2 + 2\eta \sum_{j=0}^{i-1} \langle z_j, x_j - x_t^\star \rangle (1-\lambda)^{i-1-j}$$

$$+ 2\eta^2 \sum_{j=0}^{i-1} \|z_j\|^2 (1-\lambda)^{i-1-j} + \frac{\eta}{\mu} \sum_{j=0}^{i-1} G_{j,t}^2 (1-\lambda)^{i-1-j}$$

for all $i \geq 1$. Therefore

$$\sum_{i=0}^{t-1} \|x_i - x_t^\star\|^2 (1-\lambda)^{2(t-1-i)}$$

$$\leq \|x_0 - x_t^\star\|^2 \sum_{i=0}^{t-1} (1-\lambda)^{2(t-1)-i} + 2\eta \sum_{i=1}^{t-1} \sum_{j=0}^{i-1} \langle z_j, x_j - x_t^\star \rangle (1-\lambda)^{2t-3-j-i}$$

$$+ 2\eta^2 \sum_{i=1}^{t-1} \sum_{j=0}^{i-1} \|z_j\|^2 (1-\lambda)^{2t-3-j-i} + \frac{\eta}{\mu} \sum_{i=1}^{t-1} \sum_{j=0}^{i-1} G_{j,t}^2 (1-\lambda)^{2t-3-j-i}.$$

Now we compute

$$\sum_{i=0}^{t-1} (1-\lambda)^{2(t-1)-i} = (1-\lambda)^{t-1} \sum_{i=0}^{t-1} (1-\lambda)^{t-1-i} < \tfrac{1}{\lambda} (1-\lambda)^{t-1}$$

and observe that for any scalar sequence $(X_j)_{j=0}^{t-2}$, we have

$$\sum_{i=1}^{t-1} \sum_{j=0}^{i-1} X_j (1-\lambda)^{2t-3-j-i} = \sum_{j=0}^{t-2} \left( \sum_{i=j+1}^{t-1} (1-\lambda)^{t-2-i} \right) X_j (1-\lambda)^{t-1-j}.$$

Further, if $X_j \geq 0$ for all $j = 0, \ldots, t-2$, then we have

$$\sum_{i=1}^{t-1} \sum_{j=0}^{i-1} X_j (1-\lambda)^{2t-3-j-i} = \sum_{j=0}^{t-2} \left( \sum_{i=j+1}^{t-1} (1-\lambda)^{t-1-i} \right) X_j (1-\lambda)^{t-2-j}$$

$$\leq \tfrac{1}{\lambda} \sum_{j=0}^{t-2} X_j (1-\lambda)^{t-2-j}.$$

Hence the following estimation holds:

$$\sum_{i=0}^{t-1} \|x_i - x_t^\star\|^2 (1-\lambda)^{2(t-1-i)} \leq \sum_{j=0}^{t-2} \left( 2\eta \sum_{i=j+1}^{t-1} (1-\lambda)^{t-2-i} \right) \langle z_j, x_j - x_t^\star \rangle (1-\lambda)^{t-1-j}$$

$$+ \tfrac{1}{\lambda} (1-\lambda)^{t-1} \|x_0 - x_t^\star\|^2 + \tfrac{2\eta^2}{\lambda} \sum_{j=0}^{t-2} \|z_j\|^2 (1-\lambda)^{t-2-j}$$

$$+ \tfrac{\eta}{\mu\lambda} \sum_{j=0}^{t-2} G_{j,t}^2 (1-\lambda)^{t-2-j}.$$

This completes the proof.

## A.5 Proof of Theorem 5.19

For each index $k$, let $t_k := T_0 + \cdots + T_{k-1}$ (with $t_0 := 0$) and $G_k := \eta_k \sigma^2 + \Delta^2/\mu\eta_\star^2$. Then taking into account $\eta_k \geq \eta_\star$ and our selection of the absolute constant $c$ via (3), it follows that for all indices $k$ the estimate

$$\varphi_{t_{k+1}}(X_{k+1}) - \varphi_{t_{k+1}}^\star \leq c\left(\left(1 - \frac{\mu\eta_k}{2}\right)^{T_k}\left(\varphi_{t_k}(X_k) - \varphi_{t_k}^\star\right) + \eta_k\sigma^2 + \frac{\Delta^2}{\mu\eta_k^2}\right)\log\left(\frac{e}{\delta}\right)$$

$$\leq c\left(e^{-\mu\eta_k T_k/2}\left(\varphi_{t_k}(X_k) - \varphi_{t_k}^\star\right) + G_k\right)\log\left(\frac{e}{\delta}\right)$$

holds with probability at least $1 - \delta$.

We will verify by induction that for all indices $k \geq 1$, the estimate

$$\varphi_{t_k}(X_k) - \varphi_{t_k}^\star \leq 3cG_{k-1}\log\left(\frac{e}{\delta}\right)$$

holds with probability at least $1 - k\delta$. To see the base case, observe that the estimate

$$\varphi_{t_1}(X_1) - \varphi_{t_1}^\star \leq c\left(e^{-\mu\eta_0 T_0/2}(\varphi_0(x_0) - \varphi_0^\star) + G_0\right)\log\left(\frac{e}{\delta}\right) \leq 3cG_0\log\left(\frac{e}{\delta}\right)$$

holds with probability at least $1 - \delta$. Now assume the claim holds for some index $k \geq 1$. Then because we also have

$$\varphi_{t_{k+1}}(X_{k+1}) - \varphi_{t_{k+1}}^\star \leq c\left(e^{-\mu\eta_k T_k/2}\left(\varphi_{t_k}(X_k) - \varphi_{t_k}^\star\right) + G_k\right)\log\left(\frac{e}{\delta}\right)$$

$$\leq c\left(\frac{1}{4c\log(e/\delta)}\left(\varphi_{t_k}(X_k) - \varphi_{t_k}^\star\right) + G_k\right)\log\left(\frac{e}{\delta}\right)$$

$$\leq c\left(\frac{G_k}{2cG_{k-1}\log(e/\delta)}\left(\varphi_{t_k}(X_k) - \varphi_{t_k}^\star\right) + G_k\right)\log\left(\frac{e}{\delta}\right)$$

with probability at least $1 - \delta$, a union bound reveals that the estimate

$$\varphi_{t_{k+1}}(X_{k+1}) - \varphi_{t_{k+1}}^\star \leq 3cG_k\log\left(\frac{e}{\delta}\right)$$

holds with probability at least $1 - (k+1)\delta$, thereby completing the induction. In particular, $\varphi_T(X_K) - \varphi_T^\star \leq 3cG_{K-1}\log(e/\delta)$ with probability at least $1 - K\delta$.

Next, observe

$$G_{K-1} - \sqrt[3]{\frac{27}{4}} \cdot \mu\left(\frac{\Delta\sigma^2}{\mu^2}\right)^{2/3} = \sigma^2(\eta_{K-1} - \eta_\star) = \sigma^2 \cdot \frac{\eta_0 - \eta_\star}{2^{K-1}} \leq \frac{\mu}{2}\left(\frac{\Delta\sigma^2}{\mu^2}\right)^{2/3},$$

so

$$\varphi_T(X_K) - \varphi_T^\star \leq 3c\left(\tfrac{1}{2} + \sqrt[3]{\tfrac{27}{4}}\right) \cdot \mu\left(\frac{\Delta\sigma^2}{\mu^2}\right)^{2/3}\log\left(\frac{e}{\delta}\right) \asymp \mathcal{G}\log\left(\frac{e}{\delta}\right)$$

with probability at least $1 - K\delta$. Finally, note

$$T \lesssim \frac{L}{\mu}\log\left(\frac{LD}{\sigma^2}\right)^+ + \left(1 \vee \log\log\frac{e}{\delta}\right)\frac{1}{\mu}\sum_{k=1}^{K-1}\frac{1}{\eta_k}$$

and

$$\sum_{k=1}^{K-1}\frac{1}{\eta_k} \leq 2L\sum_{k=1}^{K-1}2^k \leq 2L \cdot 2^K = 8L \cdot 2^{K-2} \leq 8\left(\frac{\sigma^2\mu}{\Delta^2}\right)^{1/3} = 8\sigma^2 \cdot \mu^{-1}\left(\frac{\Delta\sigma^2}{\mu^2}\right)^{-2/3} \asymp \frac{\sigma^2}{\mathcal{G}}.$$

This completes the proof.