# OpenReview forum: "Stochastic optimization under time drift: iterate averaging, step-decay schedules, and high probability guarantees"
_NeurIPS.cc/2021/Conference — NeurIPS 2021 Poster_

### Official Review · Reviewer_wsFZ · 2021-07-12

**Rating:** 7
**Confidence:** 3

**Summary:**

This paper studies stochastic optimization under non-stationary drift. Losses are assumed
strongly convex with Lipschitz gradients and unbounded domain. Finite-sample
bounds are derived for the tracking errors
$||x_t - x_t^*||$ and $\ell_t(x_t)-\min_{x^*}\ell_t(x^*)$ of SGD, both in-expectation and with high-probability.


**Limitations And Societal Impact:**

Yes.

**Main Review:**

Overall I thought the paper was excellent. The analysis is useful and interesting, and the writing
was clear and precise throughout. I could
see it becoming a go-to reference for myself, and I can think of several colleagues who would be
equally enthusiastic.
My questions and concerns are mostly minor, revolving around
adaptive filtering and presentation.

**Adaptive Filtering**
First, I'm generally a bit suspicious about how much the results presented here actually
relate to adaptive filtering, as is claimed. It seems to me that the results relate only
to filtering --- the optimal choice of $\eta$ depends on knowledge of $\Delta$, which would
amount to knowing the variance of the process noise driving the hidden state. Concretely, in the classic
least-squares setting, we'd have $x_{t}^*=F_{t} x_{t-1}^* +u_t$, $y_t = H_t x_t^* + v_t$ , where $u_t\sim\mathcal{N}(0, Q)$ and $v_t\sim N(0, R)$, and
we want to track $x_t^*$ through observations of $y_t$. The results presented here seem to suggest we'd need to
know both $R$ and $Q$ in order to set $\eta=\eta^*$, but given this information there's really nothing adaptive about the problem ---
not knowing these variances is precisely the difficulty of adaptive filtering!
However, I think a lot of the discussion surrounding Adaptive Filtering would be just as effective rephrased in terms
of regular filtering. I would also be curious to see exactly how the iterates chosen when using the optimal fixed $\eta^*$
relate to the steady-state solution of the Kalman filter, which *also* requires knowledge of the noise variances to
set its step-size/gain matrix.

Previous point aside, it seems like the connection to filtering could only hold
in the 1d and/or time-invariant case without additional information? In higher dimensions, even the
squared loss $\frac{1}{2}(y_t - F_tx_t)^2$ will only satisfy the strong-convexity assumption in a rather small subset
of filtering problems.
In my (quite limited) experience with filtering, it seemed like additional
assumptions on the transition/measurement matrices (e.g. the observability matrix is invertable, etc)
to say much of anything in higher dimensions.

**Presentation**
I am quite irritated with the supplementary materials. Rather than an appendix containing
technical details, the supplementary materials are an extended version of the paper,
also with its own appendix. This feels like a rather dishonest way to get around the
page limit requirements. It was also just generally annoying to navigate, and find what
I was looking for, with some results now being mixed in along side the things I already read
in the main paper, some being in the appendix of the supplementary, no real indication of
where I should look to find something mentioned in the main text. For now I've adopted the
optimistic view that this was a product of time restrictions and a properly structured
appendix is in the works.

The experiments were the weakest part of the paper to me; in my opinion these
could be moved entirely to the appendix to make room for some of the other
results. Further, there's actually a 9 page limit this year, so there's
potentially two full pages that could be used to avoid deferring so many of the results to the appendix.


**Time Spent Reviewing:**

Unsure. Reviewed with another paper over the course of a weekend. Periodic distraction by interesting citations.

---

> ### Author Response · Authors · 2021-08-10
> **Response to Reviewer wsFZ**
>
> Thank you for your complimentary remarks. We are happy that you enjoyed the paper and that you will consider using it as a reference on the topic.
>
> 1)  Adaptive filtering vs. filtering
>
> Thank you for pointing this out. We will adjust the language to “regular filtering” as you suggest.
>
> 2)  Steady state of the Kalman Filter
>
> The framework we present here is closer to the LMS algorithm than to the Kalman filter algorithm. Our framework allows us to consider the regularized convex learning objectives, common in supervised classification or regression. In terms of theoretical results, compared to the filtering literature (e.g. [13]) where the theoretical bounds are mean-square bounds, our bounds are non-asymptotic high-probability bounds. Moreover, the ‘steady-state behavior’ is studied for specific families of stochastic drifts such as Brownian motion with linear drift, whereas our analysis focuses on general conditions on the time drift (Assumption 3.1.1, Assumption 3.4.1).
>
> 3)  Connection to filtering
>
> The state space in classical filtering is indeed usually small and is typically strongly convex. In higher dimensions, one can always incorporate quadratic regularization.
>
> 4)  Presentation
>
> Thank you for pointing this out. This was indeed a product of time restrictions. As you insightfully suggest, we will move the experiments to the appendix, thereby leaving more room in the main text to discuss the results and proof techniques in greater detail. We will also add pointers in the main text to the precise locations of detailed proofs in the appendix, so as to ease navigation.

---

> > ### Comment · Reviewer_wsFZ · 2021-08-14
> > **Reviewer wsFZ's Response to the Author's Response to Reviewer wsFZ**
> >
> > Thank you for the detailed reply. Overall I am convinced by the author feedback to each of the reviews, and retain my confidence that this is an excellent paper.
> >
> > I have 1 clarification to the above
> > - My point in 2. is that the Kalman filter reduces to an LMS algorithm once it reaches its steady state. That is, there is a transient initial phase in which the filter has no (simple) closed-form, but when the filter reaches its steady-state what you're left with is a recursive least-squares estimator with some fixed gain matrix/step-size. I was just wondering if the step-size that would be prescribed by your theory matches the steady-state KF if you were to apply your results in a standard filtering problem. This would be quite interesting if it did match since it would strongly support the connection to filtering, as it would demonstrate that your theory naturally captures the known statistically optimal algorithm. So in point 2 I was not calling novelty into question, but rather suggesting that there *might* be an untapped additional argument in favor of your analysis

---

> > > ### Author Response · Authors · 2021-08-17
> > > **Steady-State KF**
> > >
> > > Thank you for your clarification - the comparison to optimal step size in steady-state KF is intriguing. It is not clear to us at this time how to align classical work on KF convergence asymptotics with our non-asymptotic convergence results on PSGD under time drift.

---

### Official Review · Reviewer_6E1U · 2021-07-13

**Rating:** 6
**Confidence:** 5

**Summary:**

the paper deals with tracking the minimum of a convex function which is evolving in time according to an unknown stochastic dynamic. The racking problem is classical in signal processing and automatic control (identification of linear or non-linear systems which are changing over time). It is less familiar in ML despite the fact that the problem of concept drift is very important in current practice.
The paper presents finite-time bound both in expectation and in high-probability for the tracjking of a a proximal stochastic gradient algorithm. The proposed result displays the relations between the learning rate, the excess risk and tracking error.

The paper uses rather stringent technical assumptions. In particular, the average loss $f_t$ are $\mu$-strongly convex, and the gradient  $\nabla f_{t}$  is  $L$-Lipschitz continuous.  The noise $z_{t}:=\nabla f_{t}\left(x_{t}\right)-g_{t}$ is a martingale increment with uniformly bounded variance.

The paper first provides finite time bounds for $\mathbb{E}\left\|x_{t}-x_{t}^{\star}\right\|^{2}$ under the assumption that $\mathbb{E}\left\|x_{t}^{\star}-x_{t+1}^{\star}\right\|^{2} \leq \Delta^{2}$. The bound in Theorem 3.2. has a transient term, a fluctuation term and a tracking error. The fluctuation term is proportional to the gradient noise variance and the stepsize. The tracking error is inversely proportional to the square of the stepsize.  This result coincides to the classical tracking error for linear system identification (under the considered stochastic model for the parameter).
The authors identify two regimes depending on the ratio  $\Delta / \sigma$.  The high drift-to-noise is not interesting (the stepsize should then be set like in the stationary case), but the low drift-to-noise  necessitates a  learning which is typicallt smaller that $1/2L$. A finite time bound is obtained in Theorem 3.2 (for constant learning rate) and Theorem 3.8 (for a step-decay schedule). In Section 3.3, the authors develop high-probability guarantees under the assumption that the tracking error is (conditionally to the past) sub-exponential and the gradient noise is sub-gaussian. The HP bound (for fixed stepsize) is given in Theorem 3.10.



**Main Review:**

Originality: To my knowledge, this is the first paper that deals with tracking in machine learning. The study of tracking is very classical in system theory, but not in ML. Most of the work in pursuit is devoted to the identification of linear systems. The work on bounds in HP is also new.

Quality:  The paper is well written and the main arguments are easy to follow. The proofs are  adapted from the classical arguments in the stationary case, so there are no really major surprises.  I would have liked a more detailed discussion of the assumptions. The assumption of sub-Gaussianity of the gradient seems to me to be costly. If the $f_t(x)= \mathbb{E}[ (y - \varphi^x)^2]$ where $\varphi \sim N(0,\sigma^2 Id) $ (the "easy" gaussian linear regression), and if $\tilde{\nabla} f_t(x)= \varphi_t ( y_t - \varphi_t x)$ (assuming for simplicty $\{(y_t,\varphi_t)\}$ i.i.d. then this assumption is not satisfied. Even if the regression vector is  bounded, then the assumption is not satisfied (unless $x_t$ is bounded). Therefore the assumption on the noise term seems strong. It would be good to check the assumption whether the condition is satisfied for the $\ell_2^2$-regularized logisitc regression (under which condition on the regressors ?). When reading the example, it is not clear to understand what you mean by "using a noisy gradient $g_t$ satisfying $\left\|g_{t}-\nabla f_{t}\left(x_{t}\right)\right\| \leq \sigma$ for each step  $t$



Significance: This is a essentially theoretical paper, a  first step in analyzing the tracking ability of stochastic gradient algorithms. Of course, this work opens many research directions and questions. It is clear that all the results remain very theoretical, one would like to have algorithms able to adapt to different constants (lipschitz, strong convexity, non-stationarity). The assumptions are very strong (in particular, it is very restrictive to assume a priori that the gradient noise is sub-gaussian, this precludes the use of this result even in very simple examples).

**Time Spent Reviewing:**

3

---

> ### Author Response · Authors · 2021-08-10
> **Response to Reviewer 6E1U**
>
> Thank you for your complimentary remarks.
>
> 1)  The paper uses rather stringent technical assumptions.
>
> The assumptions we use are in line with standard assumptions in stochastic optimization [15, 23] as well as the paper “Robust stochastic approximation approach to stochastic programming”, Nemirovski et al. (2009). In particular, the bounds in expectation do not require any light tail assumptions, and rely only on moment bounds. We certainly agree that the high probability guarantees, as stated, do not apply to your specific example. The main applications of the high probability results, as stated, are in cases of bounded noise and bounded domains. We will add a discussion to this effect in the final version of the paper. We assumed sub-Gaussianity for simplicity of exposition and to be able to apply the generalized Freedman inequality; this assumption is also in line with recent work in stochastic optimization (e.g. [15] actually assumes bounded noise). We certainly agree it is possible to consider gradient noise with heavier tails using Orlicz norms (in particular for distance tracking), or to modify the algorithm by clipping the gradient so as to handle heavy tailed noise. This is an interesting direction for a longer journal paper, but would have made the conference paper much more terse.
>
> 2)  Noise in logistic regression.
>
> Thank you for pointing out the important case of noise in logistic regression. This is indeed a very good example where the gradient noise is sub-Gaussian under a sub-Gaussian feature model; we will refer to this example in the paper. The gradient of the $\ell_2^2$-regularized logistic regression objective $$\sum_{i=1}^{n}\log(1 + \exp\langle a_i, x\rangle) - \langle Ax,b \rangle + \frac{1}{2}\|x\|^2 $$ is $$\widetilde{\nabla}f(x)=A^T(\sigma(Ax)-b)+x,$$ where the matrix $A\in\mathbb{R}^{n\times d}$ has rows $a_1,...,a_n\in\mathbb{R}^d$ and $\sigma$ denotes the sigmoid function. Being that the sigmoid function and the label vector $b$ are bounded, it therefore follows that if the rows of $A$ are sub-Gaussian, then so is the gradient noise $${\nabla}f(x) - \widetilde{\nabla}f(x) = \mathbb{E}_{A,b}\big[A^T(\sigma(Ax)-b)\big] -  A^T(\sigma(Ax)-b).$$
>
> 3)  “using a noisy gradient g_t satisfying…”
>
> In that experiment, we added a uniformly bounded noise to the true gradient as an illustration. In the final version, we will present results for an experiment where the noise in the gradient is due to using the gradient of one of the summands.

---

> > ### Comment · Reviewer_6E1U · 2021-08-11
> > **Noise condition**
> >
> >  I am fully convinced by the answer given. The  noise conditions are indeed restrictive, but they are very commonly used in the literature, as the authors rightly point out

---

### Official Review · Reviewer_YZ6G · 2021-07-14

**Rating:** 7
**Confidence:** 3

**Summary:**

This work considers the problem of stochastic gradient descent for time varying optimization. Based on the analysis of SGD in this setting, the problem actually contains two qualitatively different regimes: one with high drift-to-noise ratio, and one with low drift-to-noise ratio. The paper further provides tight convergence analysis (both in expectation and in high probability) for the two regimes.

**Limitations And Societal Impact:**

The potential negative societal impact is addressed properly.

I don't have any major problem for this paper. There are only some minor problems (the numbering of the lines is for the supplementary material):

The checklist is unfinished. The authors should do a double-check.

The figures do not show the "95% CI" indicated in the legend.

Line 323, one of the "Theorem 4.2" should be "Theorem 3.2".

**Main Review:**

The paper gives a new way of looking at time varting optimization. The proof techniques are not new, but the discovery of the two regimes is very interesting and this implication can be useful for real world applications.

This paper is clearly written and well-organized.

**Time Spent Reviewing:**

4

---

> ### Author Response · Authors · 2021-08-10
> **Response to Reviewer YZ6G**
>
> Thank you for your complimentary remarks. We are glad you enjoyed the paper.
>
> 1)  Checklist
>
> We managed to complete the checklist by the deadline for submitting the supplementary document and it appears there.
>
> 2)  “95% CI”
>
> Thank you for catching this. In the experiment, the “95% CI” are so narrow that they are barely visible. We will try to make them more visible in the final version.
>
> 3)  Line 323
>
> Thank you for catching this. We will fix the typo in the final version.

---

> > ### Comment · Reviewer_YZ6G · 2021-09-01
> > **Reponse to the authors**
> >
> > Thanks for the response. I intend to keep my current score and suggest to accept the paper.

---

### Official Review · Reviewer_WvZE · 2021-07-16

**Rating:** 6
**Confidence:** 4

**Summary:**

The paper under review is devoted to the analysis of the stochastic approximation algorithm with the presence of the drift term. The main result is to derive finite-time bounds on the expected mean square error.

**Limitations And Societal Impact:**

The work has no potential negative impact.

**Main Review:**

The authors consider the problem of form
$
min f(x) + r_t(x)
$, where $r_t$ is a drift term. The author assumes that the stochastic gradient used for solving the problem is unbiased. Under that assumptions, the author proves finite-time bounds. The results seem to be correct but I have serious doubts about novelty.  The assumptions about bias and assumptions on drift term in my opinion make that such a bound are well known in the community. In my opinion, this kind of result can be straightforwardly obtained from Athchade et al 2016 or Rossaco et al 2014. Due to that, I do not believe that the paper meets the standards of NIPS conference.

####UPDATE
After reading the author response and other reviews, I agree that  I underestimate the novelty of the paper.


**Time Spent Reviewing:**

10h

---

> ### Author Response · Authors · 2021-08-10
> **Response to Reviewer WvZE**
>
> 1)  The reviewer summarizes the paper as considering "…the problem min_x f(x)+r_t(x), where r_t is a drift term."
>
> To clarify, r_t is a regularizer here (typically independent of time) and it is the loss/risk f_t that is varying in time. We are thus mostly focusing on the problem min_x f_t(x)+r(x).
>
> 2)  “The assumptions about bias and assumptions on drift term in my opinion make that such a bound are well known in the community.”
>
> The main results in our paper are new; we are not aware of any published paper that establishes the finite-time bounds stated in Theorems 2.7, 3.2, 3.3, 3.5, and 3.6. Using the step-decay schedule for obtaining logarithmic dependence on initialization is an important contribution of the work, which again we have not seen in the available literature. The high probability guarantees build off the generalized Freedman inequality, which was only recently proved by Harvey et al. in COLT 2019 [18].
>
> 3)  “In my opinion, this kind of result can be straightforwardly obtained from Athchade et al 2016 or Rossaco et al 2014.”
>
> We searched for these references given the limited information provided by the reviewer. The closest references we could find were “On perturbed proximal gradient algorithms” by Atchade, Fort, and Moulines (2014), and “Learning with incremental iterative regularization” by Rosasco, Villa (2014). We comment on these in turn. Atchade et al. consider a static problem (no time drift) and stochastic gradients obtained via Markov Chain Monte Carlo, resulting in biased gradients. We fail to see how our results relate to theirs except at a superficial level. Rosasco et al. consider a static problem (no time drift) and show that early stopping acts as a regularization on the learned parameters for a least squares objective. Again, we do not see any relevance of this paper to our results.
>
> We argue that our results are far from straightforward, relying on very recently developed techniques in stochastic optimization (e.g. step decay, careful iterate averaging) and concentration inequalities for self-regulating stochastic processes (e.g. SGD iterates).

---

### Decision · Program_Chairs · 2021-09-27

**Decision:**

Accept (Poster)

**Comment:**

All the reviewers were positive about this paper and felt that it presented a good analysis of less-well-studied problem.